# A Novel *Enterococcus*-Based Nanofertilizer Promotes Seedling Growth and Vigor in Wheat (*Triticum aestivum* L.)

**DOI:** 10.3390/plants13202875

**Published:** 2024-10-14

**Authors:** Salma Batool, Maryam Safdar, Saira Naseem, Abdul Sami, Rahman Shah Zaib Saleem, Estíbaliz Larrainzar, Izzah Shahid

**Affiliations:** 1Department of Basic and Applied Chemistry, Faculty of Science and Technology, University of Central Punjab, 1-Khayaban-e-Jinnah Road, Johar Town, Lahore 54782, Pakistan; salma.batool@ucp.edu.pk (S.B.); maryamsafdar439@gmail.com (M.S.); 2Department of Biotechnology, Faculty of Science and Technology, University of Central Punjab, 1-Khayaban-e-Jinnah Road, Johar Town, Lahore 54782, Pakistan; sairanaseem1@gmail.com; 3H.A. Shah & Sons Group of Companies, Islamabad 46000, Pakistan; abdulsami2005@gmail.com; 4Department of Chemistry and Chemical Engineering, SBA School of Science and Engineering (SBASSE), Lahore University of Management Sciences (LUMS), Lahore 54792, Pakistan; rahman.saleem@lums.edu.pk; 5Institute of Multidisciplinary Research in Applied Biology (IMAB), Public University of Navarre (UPNA), Campus Arrosadia, 31006 Pamplona, Spain; estibaliz.larrainzar@unavarra.es

**Keywords:** biofertilizers, green synthesis, nanoparticles, plant growth promoting bacteria, silver, wheat

## Abstract

Excessive use of chemical fertilizers poses significant environmental and health concerns. Microbial-based biofertilizers are increasingly being promoted as safe alternatives. However, they have limitations such as gaining farmers’ trust, the need for technical expertise, and the variable performance of microbes in the field. The development of nanobiofertilizers as agro-stimulants and agro-protective agents for climate-smart and sustainable agriculture could overcome these limitations. In the present study, auxin-producing *Enterococcus* sp. SR9, based on its plant growth-promoting traits, was selected for the microbe-assisted synthesis of silver nanoparticles (AgNPs). These microbial-nanoparticles SR9AgNPs were characterized using UV/Vis spectrophotometry, scanning electron microscopy, and a size analyzer. To test the efficacy of SR9AgNPs compared to treatment with the SR9 isolate alone, the germination rates of cucumber (*Cucumis sativus*), tomato (*Solanum lycopersicum*), and wheat (*Triticum aestivum* L.) seeds were analyzed. The data revealed that seeds simultaneously treated with SR9AgNPs and SR9 showed better germination rates than untreated control plants. In the case of vigor, wheat showed the most positive response to the nanoparticle treatment, with a higher vigor index than the other crops analyzed. The toxicity assessment of SR9AgNPs demonstrated no apparent toxicity at a concentration of 100 ppm, resulting in the highest germination and biomass gain in wheat seedlings. This work represents the first step in the characterization of microbial-assisted SR9AgNPs and encourages future studies to extend these conclusions to other relevant crops under field conditions.

## 1. Introduction

Given the current global climate landscape, food production to feed the growing populations remains a challenge. Although intensive use of chemical fertilizers has helped meet food demands, their excessive input has been shown to cause health and environmental hazards [1,2]. Furthermore, crop nutrient deficiency remains a challenge despite regular exposure to fertilizers. The primary reasons for the unavailability of essential nutrients in the form of fertilizers include evaporation, leaching, runoff, and microbial degradation [3]. The hazards of environmental pollution and disturbance of soil microbiota due to the wide range of chemicals in fertilizers are parallel issues [4,5].

Biofertilizers have been presented as an alternative to avoid the harmful effects of chemical fertilizers and to ensure the availability of nutrients in crops. Biofertilizers are bio-based organic fertilizers derived from plant and animal sources or microbial-based fertilizers [6]. Microbial biofertilizers include plant growth-promoting rhizobacteria (PGPR), arbuscular mycorrhizal fungi (AMF), and plant-beneficial algae [7]. Enriching the soil microbiome can stimulate plant growth through mineral solubilization and mobilization, production of phytohormones and catalytic enzymes, and induction of systemic defense pathways against biotic and abiotic stressors [8,9]. Among the phytohormones produced by rhizobacteria, auxin plays a noteworthy role in promoting root growth, ensuring better overall plant health, and increased yield. Several auxin-producing bacteria from the genera *Rhizobium*, *Agrobacterium*, *Bacillus*, *Bradyrhizobium*, and *Azospirillum* have been shown to improve the growth of radish, lettuce, corn, rice, spinach, barley, and wheat [10,11]. Unfortunately, when applied under field conditions, plant growth-promoting bacteria (PGPB)-based biofertilizers often fail to deliver desirable field results; hence, their efficacy remains the biggest concern in their use. For instance, in a consortium, microorganisms exhibit diverse growth requirements and may not synchronize with other consortium members. This phenomenon can result in a decrease in the bacterial cell population within the soil, which is insufficient for the effective colonization of the rhizosphere to confer beneficial effects on plant growth [12]. Furthermore, biofertilizers containing extracellular polymeric substance (EPS)-producing PGPB are considered efficient rhizosphere colonizers because of their biofilm formation ability. Nonetheless, several critical regulatory factors, including environmental and soil conditions, modulate biofilm formation by EPS-producing PGPB in the rhizosphere. This robust competition between the soil environment and biofertilizers impedes biofilm formation capability, leading to PGPB-dysfunctional bioinocula [13].

A substitute approach based on the development of nanobiofertilizers can overcome some of the limitations of bacterial biofertilizers, thereby increasing their efficacy and shelf-life. Microbe-mediated synthesis of nanofertilizers offers a controlled and targeted release of nutrients into the soil, ensuring potential plant growth stimulation [14]. Nanobiofertilizers can also improve soil quality by enriching the rhizomicrobiome, leading to increased soil fertility and crop productivity [15,16,17]. Nanobiofertilizers can be designed using bottom-up and top-down approaches, as well as diverse biological methods [18]. Several biogenic platforms have been reported for the metallic synthesis of nanoformulations, such as plant extracts and bacteria. Such biogenic materials offer efficient and cost-effective manipulation in terms of manufacturing and lack toxic substances that are harmful to human health [12,13,14]. The use of microbes, particularly bacteria, to synthesize metallic nanoformulations has emerged as an innovative approach because of the tendency of microorganisms to accumulate and extract metallic nanoparticles directly in the synthesis medium [15,16,17,18]. Various bacterial biomolecules such as polysaccharides, proteins, organic acids, and reductases act as reducing and stabilizing agents to prevent nanoparticle aggregation [19]. Additionally, extracellular polymeric substances (EPS) can aid in the extracellular synthesis of NPs in the medium [20,21]. The delivery of these nanoformulations into the soil is ensured through various techniques, including aeroponic, hydroponic, and soil applications [22,23,24]. Plants’ uptake of nanoparticles occurs by binding with carrier proteins through aquaporins, ion channels, or endocytosis, resulting in plant growth-promoting effects.

Among the various nanoparticles exploited, silver nanoparticles (AgNPs) have great potential for enhancing crop production by increasing seed germination and plant growth [25]. It is important to note, however, that plants treated with nanoparticles can absorb and transfer them to various components, influencing growth and biomass depending on the dosage, size, and duration of exposure. Notably, AgNPs have both advantageous and deleterious effects on various physiological processes, including seed germination, root elongation, cellular division, chromosomal alterations, and metabolic activity. Kumar et al. [26] reported that AgNP50, with a size of 15.5–21 nm, resulted in the highest seed germination rate in *Psophocarpus tetragonolobus*. Similarly, green-synthesized AgNPs imparted phytostimulatory effects on rice seedlings by ensuring faster ATP production, higher photosynthetic pigment content, and increased respiration [27]. In contrast, Thiruvengadam et al. [28] reported conflicting findings, indicating a decreased chlorophyll content at higher concentrations of AgNPs. Nevertheless, the application of AgNPs has been proposed to lower oxidative stress by increasing the activity of antioxidant enzymes in wheat seeds in saline environments [29]. In addition to the potential use of biofertilizers, some studies have reported the fungicidal effects of AgNPs, indicating their potential role as biofungicides [30,31]. Overall, this study indicates the dominance of positive attributes over negative attributes of AgNP-based nanoformulations for agricultural purposes.

In the current study, a plant growth-promoting auxin-producing bacterium *Enterococcus* sp. SR9 was used for the synthesis of a novel silver-based nanofertilizer. We assessed the effects of microbially synthesized SR9AgNPs on seed germination and plant growth in several agronomically relevant crops. The combined potential of SR9 and SR9AgNPs was also evaluated, which has not been previously reported. The potential toxic effects of SR9AgNPs on seedlings were also evaluated, and under the tested conditions, the application of AgNPs had no negative effects in planta.

## 2. Results

### 2.1. Identification of Plant Growth-Promoting Traits in Spinach Rhizobacteria

Spinach rhizobacteria were isolated using the serial dilution method, and Petri plates containing bacterial colonies were assessed. Dense clusters and overlapping bacterial colonies were observed on plates containing dilutions 10^1^–10^2^. Petri dishes containing dilutions of 10^3^–10^6^ demonstrated a gradient pattern and morphologically distinct bacterial isolates (Figure 1A–C). Single bacterial colonies were selected from each plate and subcultured onto LB agar plates for purification. In total, 12 bacterial isolates were purified from the rhizosphere and rhizoplane regions based on the different morphologies and appearance of the colonies. All isolates were serially named SR1–SR12 (Figure 1D–I).

In terms of biochemical characterization of the isolates, most of them tested positive for o-nitrophenyl-β-D-galactopyranoside, sodium citrate, sodium malonate, arginine dihydrolase, tryptophan deaminase, and Voges-Proskauer/acetoin production, among others (Appendix A). In contrast, they did not show any activity for ornithine decarboxylase, hydrogen sulfide, or urea hydrolysis. Regarding the use of different carbon sources, most isolates could utilize multiple compounds, such as glucose, maltose, arabinose, melibiose, raffinose, sorbitol, and mannitol. Interestingly, only half of the bacterial culture grew on sucrose (Appendix A). Finally, four bacteria (SR3, SR6, SR7, and SR11) were negative for cytochrome oxidase activity, whereas the remaining bacterial isolates were positive (Appendix A, Table 1).

Next, the hydrolytic activities of the bacterial isolates were analyzed. Most tested positive for lipase, amylase, protease, and catalase activity (Table 1). For phosphate solubilization, SR9, SR10, SR11, and SR12 showed positive results in the NBRIP medium, while on Pikovskaya’s (PKV) agar plates, eight bacteria (SR1, SR2, SR5, SR6, SR7, SR8, SR9, and SR10) were positive (Table 1). The results of zinc solubilization varied depending on the salt used; however, the majority were able to solubilize zinc carbonate and sulfate. Similarly, most of the isolates showed potassium solubilization capacity (Table 1). Finally, 9 out of the 12 isolates were able to produce the auxin indole-3-acetic acid (IAA), particularly strains SR3, SR9, and SR10 (Appendix A, Table 1).

### 2.2. The Isolate SR9 Had the Most Positive Impact on Wheat Germination and Vigor

To test whether the observed plant growth-promoting bacterial traits detected in several isolates could be translated into beneficial effects in planta, a germination experiment using wheat seeds was performed. Wheat seeds were sterilized, inoculated with cultures of the five most promising candidates (isolates SR1, SR2, SR3, SR9, and SR10, selected based on their dominance of plant growth-promoting traits compared to other strains), and grown on plates under controlled conditions. Fourteen days after plating, germination rates and vigor indices were calculated. All selected isolates had a positive effect on wheat growth compared to the uninoculated and non-PGPB bacterial isolate SR12 *Aeromonas hydrophilla* (Appendix A, Appendix A). Interestingly, the maximum germination rate and vigor indices were observed in wheat seedlings inoculated with isolates SR9 (98% and 1928, respectively) and SR3 (96% and 1865, respectively). Therefore, SR9, identified as *Enterococcus* sp. based on 16S rDNA sequencing, was selected for further analysis (Appendix A; GenBank accession ID OR133233).

### 2.3. Biosynthesis of SR9-Silver Nanoparticles

To synthesize the SR9-based bionanofertilizer, the bacterial culture was supplemented with an AgNO_3_ solution. A change in color from pale yellow to black was observed in the supernatant, which served as a preliminary indication of the synthesis of AgNPs (Figure 2A). This observation was further validated by UV/Vis spectrophotometric analysis at 0–48 h. The spectra showed the best absorption with a 2 mM AgNO_3_ (approximately 340 mg/L) solution, and the maximum absorption was obtained at 400–450 nm (Figure 2B). The presence of functional groups on SR9AgNPs was verified using Fourier-transform Infrared Spectroscopy (FTIR) (Figure 2C). The -OH bond was indicated at 3050.30 cm^−1^, 3016.44 cm^−1^, 2959.14 cm^−1^, 2936.28 cm^−1^ and 29.1550 cm^−1^. At 1731.79 cm^−1^, the carboxylate C=O bond was observed. The detection of a carboxylic group in the AgNPs indicated the presence of auxin indole-acetic acid in the bacterial supernatant, which may act as a capping agent (Figure 2C). In terms of characterization of the nanoparticles, the zeta sizer results indicated that they presented an average of 110.2 nm in size (Figure 3A), and the zeta potential was measured at −23.6 mV with a polydispersity index (PDI) of 0.2 (Figure 3B). These results suggest that the nanoparticles are within the recommended range, with PDI < 0.3 considered acceptable [32]. Using SEM-EDX, the surface morphology of the SR9AgNPs was found to be mostly spherical (Figure 3C), and EDX analysis showed the presence of several elements on their surface in addition to silver, including C, O, Na, Au, and S (Figure 3D), which have been shown to have a positive effect on bio-nanofertilizers [33,34].

### 2.4. Cytotoxicity Assessment and In-Planta Effect of SR9AgNPs

AgNPs have been described to have both positive and negative effects on plant growth depending on their concentration. Therefore, before its application as a nanofertilizer, the potential cytotoxicity of SR9AgNPs at various Ag concentrations in wheat seeds was tested. Seeds were treated with different concentrations of SR9AgNPs and germinated on plates under controlled conditions. After 14 days of growth, the petri plates were removed from the growth chamber, and the harvested seedlings were analyzed to determine the optimal non-cytotoxic concentration of SR9AgNP. Wheat seedlings coated with 100 ppm SR9AgNPs showed the highest growth and germination rates, followed by seeds coated with 50 ppm, the lowest concentration (Appendix A, Table 2). Therefore, 100 ppm was selected as the reference concentration for the nanobiofertilizers experiment.

Next, the effect of the biosynthesized SR9AgNPs was tested in three major crops: wheat, cucumber, and tomato. Three treatments were used: the bacterial isolate SR9 alone, silver nanoparticles synthesized in the SR9 culture (SR9AgNPs), and a combination of the two (SR9 + SR9AgNPs). The results showed that the three treatments had a positive impact on both the germination and vigor of the three crops tested compared to the uninoculated control seedlings (Table 3). The highest values of germination and vigor indices were observed in plants inoculated with the combination of SR9 + SR9AgNPs, followed by those inoculated with the nanofertilizer alone SR9AgNPs (Table 3, Appendix A). Regarding the effect of nanofertilizer on plant growth and biomass, the highest values were measured in plants treated with the SR9 + SR9AgNP combination, followed by the bionanoparticle treatment (Figure 4). Interestingly, this combined treatment was particularly beneficial in promoting dry biomass accumulation, with an almost two-fold increase compared to the untreated control seeds. These results are consistent for the three plant species tested (Figure 4).

### 2.5. SR9AgNPs Treatment Can Increase the Physiological Capacities of Wheat Plants

Physiological analysis of wheat plants indicated a significant increase in the total soluble sugar, phenol, and chlorophyll content of leaves compared to the controls. The highest chlorophyll, sugar, and phenolic content was observed in plants treated with the combination of SR9 + SR9AgNP, followed by the control and *Enterococcus* SR9-treated plants (Figure 5). Nonetheless, the total protein content was significantly higher in plants treated with SR9 than those inoculated with SR9AgNPs. However, the total protein levels were still higher than those of the untreated controls. The DPPH assay indicated the capacity of SR9AgNPs to act as free-radical scavengers, thus reducing the negative impact of free radicals at the metabolic and cellular levels. DPPH serves as a stabilized free radical that is converted to the DPPHH form in the presence of an antioxidant radical scavenger. The antioxidant capacity of SR9AgNPs was notably higher than that of the control plants and the biofertilizer treatment. The IC_50_ values indicated the potential of SR9AgNPs to scavenge the free radicals generated within plants and protect the plants from reactive oxygen species (Figure 6).

## 3. Discussion

In this study, an auxin-producing *Enterococcus* sp., SR9, was selected as a promising PGPB isolate and used for microbial-assisted synthesis of a silver nanofertilizer. Several studies have analyzed the positive effects of *Enterococcus* spp. on different research dimensions. For example, *Enterococcus* sp. isolates have been used for the biosynthesis of nanoparticles with antibacterial properties and wastewater treatment [35,36]. Similarly, there are several reports on the positive effects of AgNPs as a nanofertilizer [37,38,39]. However, to our knowledge, their combined potential as bionanofertilizers has not been addressed.

*Enterococcus* sp.-based SR9AgNPs were characterized using several methods, including UV/Vis spectrophotometry, size distribution, surface potential analysis, and FTIR. These analyses showed that the nanoparticles had adequate size and potential for use as nanofertilizers [31,32,40]. Tomato, wheat, and cucumber seeds treated with SR9AgNPs and a combination of SR9 and SR9AgNPs showed a 100% germination rate. In contrast, the highest germination rate in the untreated seeds was approximately 90%. The greatest increase in plant growth was observed in SR9 + SR9AgNPs inoculated plants, indicating the synergistic action of this combination on plant growth. Similar synergistic effects have been previously reported in the literature, where a substantial increase in plant growth was observed in biofertilizer- and nanofertilizer-treated plants [41]. The synergistic effect observed in the SR9 + SR9AgNPs treatment likely stems from the combined bioactivity of the *Enterococcus* SR9 biofertilizer and AgNPs. In this case, SR9 may have promoted plant hormone production and improved root development, whereas SR9AgNPs may have stimulated microbial activity. To validate the potential mechanism, future studies could focus on molecular and physiological analyses, such as tracking nutrient assimilation, hormone production (auxins), and variable gene expression in plants treated with biofertilizer and nanofertilizer combinations. Additionally, monitoring microbial community changes in the rhizosphere can help to elucidate the role of microbial dynamics in this synergistic effect.

One of the concerns associated with the use of nanofertilizers is their phytotoxicity and cytotoxicity to non-target organisms. Several studies have demonstrated the phytotoxic effects of nano-formulations on plant development. For example, López-Moreno et al. [42] reported significantly reduced seed germination in tomato, cucumber, soybean, and maize when subjected to 2000 mg/L of nCeO_2_. Similarly, exposure to high concentrations of aluminum, zinc oxide, and zinc nanoparticles has shown phytotoxic effects on root development and seed germination in several crops [43,44]. However, it is worth noting that the nanoparticles used in most of these studies were synthesized using a metal-based approach. Microbial-assisted nanoparticle biosynthesis is generally considered safe and has shown less cytotoxicity and phytotoxicity than synthesis using other methodologies [45]. In the present study, we tested a range of concentrations of SR9AgNPs and observed phytotoxicity at the highest doses, while a 100-ppm concentration showed effective seedling growth and increased seedling vigor and weight, with no negative effects. Moreover, the higher chlorophyll, protein, sugar, and phenolic content of wheat plants also demonstrated the positive effects of SR9AgNPs at the physiological level. These NPs can also serve as antioxidants to scavenge the free radicals generated within plant tissues and protect plants from oxidative and free-radical-associated damage. For example, Fe_2_O_3_-based nanofertilizer application on peanut plants has been shown to enhance antioxidant enzymes compared with uninoculated plants [46]. Similarly, exogenous application of SiNPs in wheat has been reported to increase the levels of antioxidant enzymes, proline, flavonoids, and phenolics [47]. These results suggest that at the recommended concentration of 100 ppm, SR9AgNPs can be considered non-phytotoxic based on preliminary analysis and can be recommended for the development of nano-bioformulations in the future.

## 4. Materials and Methods

### 4.1. Isolation of Spinach Rhizobacteria

Spinach (*Spinacia oleracea*) root samples and rhizospheric soil were collected from agricultural fields in Gujranwala, Pakistan (32.1877° N, 74.1945° E). The samples were aseptically placed in sterile polythene bags and brought to the Biotechnology Laboratory at the University of Central Punjab (Lahore, Pakistan). The standard serial dilution method was used to isolate rhizospheric bacteria [48]. Roots with large soil clumps were cleaned, and 10 g of root-adhered soil was carefully collected in a flask containing 100 mL autoclaved 1% (*w*/*v*) peptone water. For isolation from the rhizoplane section, 10 g of roots were crushed in a sterile mortar and pestle, and 10 mL of 0.85% NaCl was added to the crushed roots. The slurry was transferred to a flask containing 100 mL sterile 1% (*w*/*v*) peptone water. Both flasks were incubated at 37 °C for 2 h in an orbital shaker and used to prepare serial dilutions from 10^0^–10^6^. Bacteria were cultivated by spreading 100 µL of each dilution separately on Lauria Bertani (LB) agar plates. Plates were incubated at 37 °C overnight, and the resulting bacterial isolates were sub-cultured for further purification.

### 4.2. Biochemical Identification of Bacterial Isolates

Biochemical characteristics of the bacterial isolates were assessed using QTS-24 bacterial identification kits (DESTO Laboratories, Karachi, Pakistan). Single colonies of pure bacterial isolates were inoculated separately into 10 mL of LB broth and placed in a shaking incubator at 37 °C overnight. Bacterial cultures were aseptically transferred to Falcon tubes and centrifuged at 2830× *g* for 10 min. After centrifugation, the pellets were collected, and supernatants were discarded. The pellets were resuspended in 6 mL sterilized saline (0.85%) by gentle vortexing. The QTS-24 kits were inoculated, and the results were interpreted according to the manufacturer’s guidelines. Cytochrome oxidase tests were performed to assess the activities of the bacterial isolates. Fresh bacterial colonies were harvested aseptically from the culture plates and rubbed onto cytochrome oxidase strips. A change in color from white to light purple or blue within a few seconds showed positive results. For the catalase test, a single bacterial colony was selected from the pure culture and placed on a glass slide using a sterile toothpick. A drop of hydrogen peroxide (H_2_O_2_) was added to a glass slide and observed for bubble production [49]. Lipase tests were performed on LB agar plates supplemented with Tween-80 (1%) according to the method described by Kumar et al. [50]. The inoculated LB plates were incubated overnight at 37 °C for 48 h, and the presence of precipitates was observed around the bacterial colonies, indicating positive test results. The starch hydrolysis assay was performed on LB agar plates supplemented with 1% starch. Bacterial cultures were spot-inoculated onto prepared agar plates and incubated at 37 °C for 48 h. Following incubation, the plates were stained with 1% Gram iodine reagent for 10 min. The reagent was discarded, and the plates were checked. The yellow zones around bacterial growth were considered positive [51]. Protease tests were performed on skim milk agar plates. Briefly, pure bacterial cultures were individually inoculated onto skimmed milk agar plates and incubated for 24 h at 37 °C. Clear zones around the bacterial colonies were indicative of positive test results [52].

### 4.3. Solubilization of Insoluble Minerals

The National Botanical Research Institute’s phosphate growth medium (NBRIP) and Pikovskya’s (PVK) agar medium were used to detect the solubilization ability of insoluble tricalcium phosphate [53]. Pure bacterial cultures were spot inoculated on NBRIP and PVK-agar plates and incubated at 37 °C for 14 days. The appearance of clear halo zones surrounding bacterial colonies was interpreted as the ability to solubilize phosphate. Zinc solubilization was performed on Tris minimal agar medium, individually supplemented with 0.1% zinc salts, that is, zinc carbonate and zinc sulfate [54]. Bacterial isolates were spot-inoculated onto prepared agar plates, wrapped in aluminum foil, and incubated at 37 °C for 14 d. The appearance of halo zones around the bacterial colonies was considered a positive test result. The potassium solubilization ability of the bacterial isolates was tested against KCl, KNO_2_, and KH_2_PO_4_. Aleksandrow agar medium was used to spot inoculate bacterial cultures in triplicate, and plates were incubated at 37 °C for 72 h [55]. Positive results were indicated by the formation of halo zones around bacterial growth. The solubilization index (SI) was calculated using the following formula:(1)SI = colony diameter + halo zone diameter/colony diameter

### 4.4. Evaluation of Indole-3-Acetic Acid Production

Indole-3-acetic acid production was determined using a colorimetric assay [56]. L-tryptophan (0.01%) supplemented LB broth (10 mL) was used to inoculate pure bacterial isolates as previously recommended [57,58]. Cultures were grown for seven days at 37 °C. Following incubation, 1 mL of each bacterial culture was individually transferred to clean Eppendorf tubes and centrifuged at 4464× *g* for 10 min. After centrifugation, the supernatant (0.5 mL) was transferred to another clean Eppendorf tube, and 1 mL of Salkowski’s reagent was added. The Eppendorf tubes were incubated for 30 min in the dark, and pink color formation was observed.

### 4.5. Impact of Plant Growth Promoting Bacteria on the Vigor Index of Wheat

Five bacterial isolates were selected for the wheat plant experiment based on the results of plant growth-promoting assays. The isolates, including SR1, SR2, SR3, SR9, and SR10, were grown individually in LB broth for 24 h at 37 °C. Cultures were harvested at 2830×*g*, and the pellets were suspended in 0.85% sterile saline. Subsequently, a 0.1% hypochlorite solution was used to sterilize the wheat seeds (variety FSD 2008) for 10 min. The seeds were washed in sterile dH_2_O 4–5 times and air dried. Dried seeds were suspended in bacterial cultures (30–32 seeds/culture) for 3–4 h and air-dried in a laminar flow cabinet. Uninoculated surface-sterilized seeds were used as a control [59]. To analyze the effect of PGPB on the vigor index (VI) of wheat, seeds were germinated on 1% dH_2_O-agar plates. Ten seeds per plate were placed on agar plates and kept for incubation at 25 °C. Shoot and root lengths were measured every 24 h, and the experiment was repeated in triplicate. Untreated seeds were used as controls. VI was calculated using the following formula [60]:(2)$$Germination\;Rate\left(\%\right)\;=\;No.of\;Seeds\;Germinated\;÷\;Total\;No.of\;Seeds\;\times \;100$$
(3)Vigor Index = %Germination × mean seedling length

### 4.6. Molecular Identification of Bacterial Isolates

A GeneJET Genomic DNA Purification Kit (Thermo Fisher Scientific, Waltham, MA, USA) was used to isolate DNA from the five bacterial isolates. The 16S rDNA gene was amplified in a 50 μL reaction mixture containing 25 μL of DreamTaq Green PCR Master Mix (2X), 4 μL of each forward and reverse primer (20 pmol), 5 μL of DNA sample (>50 ng/μL), and 12 μL of dH_2_O. Primer sequences were FGPS forward 1509–153 5′-AAG GAGGTGATCCAGCCGCA-3′, and FGPS reverse 4–281 5′-AGAGTTTGATCCTGGCTCAG-3′ [61]. PCR conditions were as follows: denaturation: 95 °C 1 min, annealing: 55 °C 60 s, extension: 72 °C 90 s, for 35 cycles. The amplified products were sequenced by Macrogen (Seoul, Republic of Korea). The resultant sequences were BLAST using the NCBI GenBank Sequence Database. Phylogenetic analyses were performed using the NCBI for Biotechnology Information BLAST platform.

### 4.7. Extracellular Synthesis and Optimization of Bacterial Silver Nanoparticles (SR9AgNO_3_)

Isolate SR9 was subjected to nanofertilizer synthesis based on its plant growth-promoting potential and traits. For nanoparticle synthesis, SR9 was inoculated in 1 L LB broth and aerobically cultivated for 48 h at 37 °C in an orbital shaker. The culture was centrifuged at 2830× *g* for 10 min and filtered through a Whatman No. 1 filter paper. The supernatant was divided into four equal parts (250 mL), and different concentrations of AgNO_3_ (1, 2, 5, and 10 mM) were added separately. Flasks were incubated at 37 °C and observed for colorimetric changes indicative of nanoparticle synthesis [32].

### 4.8. Characterization of SR9AgNPs

For UV–Vis absorbance spectroscopy analysis, the reaction mixture of the SR9 supernatant and AgNO_3_ was monitored for the change in color in relation to the spectrophotometric assessment of absorbance, which was recorded at 300–700 nm. Distilled water was used as a blank. The zeta potential of SR9AgNPs was assessed using a zeta sizer (LTD, ver. 7.10, Malvern, Worcestershire, UK). The sample (1 µg/mL SR9AgNPs) was resuspended and sonicated for 30 min prior to use. The software was set to automatic mode, and the hydrodynamic diameter and polydispersity index (PDI) were measured. All measurements were performed in triplicate at room temperature [62]. For Fourier transform infrared (FTIR) spectroscopy, 5 mg of dried SR9AgNP was subjected to an FTIR sample collector (ATR platinum Diamond 1 Refl) for spectrum evaluation under laboratory light in the wavelength range 400–4000 cm^−1^. Graphical data were recorded to determine the capping agents associated with the SR9AgNPs. To assess the morphology of the synthesized nanoparticles, 5 mg of the prepared SR9AgNPs was placed on a carbon-coated copper plate using a scanning electron microscopy EDX detector (SEM-EDX). After spreading the sample over the holder, the instrument (Nova Nano SEM) was used to observe the scans at several magnifications. The images and data were recorded using a computer attached to the SEM machine.

### 4.9. Cytotoxicity Assessment of SR9AgNPs

The cytotoxicity of SR9AgNPs was assessed in wheat plants. Different concentrations of SR9AgNP (50, 100, 200, 500, and 1000 ppm) were prepared in sterile dH_2_O. Wheat seeds were surface-sterilized as described above and soaked in the respective SR9AgNP dilution for 15 min. After soaking, the seeds were air-dried in a laminar cabinet and placed on 1% water-agar plates. Petri plates were incubated for 7 days, and the overall growth of seedlings against differential SR9AgNPs exposure [63].

### 4.10. Plant Experiments Using SR9AgNPs as Nanofertilizer

Seeds of wheat, cucumber, and tomato were soaked in 0.1% NaClO solution for 15 min, washed in sterile dH_2_O 3–4 times, and air-dried on sterile Whatman No. 1 filter paper. Un-inoculated surface sterilized seeds were used as the negative controls. *Enterococcus* sp. SR9 strain was overnight grown in 10 mL LB broth at 37 °C following centrifugation. The supernatant was discarded, and the cells were resuspended to a final concentration of 1 × 10^7^ CFU containing 0.025 g sucrose. Surface-sterilized tomato, cucumber, and wheat seeds were separately soaked in the bacterial culture for 30 min and air-dried under laminar flow. The treated seeds were used to evaluate the biofertilizer potential of SR9. Seed treatment with SR9AgNPs was performed by preparing 2 mL of the 100-ppm solution in a Falcon tube. Sterilized seeds were transferred to Falcon tubes for 30 min and air-dried in a laminar flow hood. To check the combined effect of SR9 and SR9AgNPs, seeds were separately soaked in 100 ppm (0.5 mL) of the synthesized SR9AgNPs + 0.5 mL of 1 × 10^7^ CFU SR9 solution for 30 min. The seeds were air-dried and used in subsequent experiments. All experiments were conducted in triplicate.

Ten seeds from each treatment group were transferred onto 1% water and agar plates. Three plates were prepared for each treatment and three for the un-inoculated controls. Plates were incubated at 23–25 °C for 14 days. Vigor index and germination percentage were calculated using the formulas described above. Plates were incubated for 14 days, and subsequent harvesting, shoot and root lengths, and plant biomass were determined. Plants were dried overnight in a hot-air oven at 60 °C for the calculation of dry plant biomass.

### 4.11. Physiological Capacities of SR9AgNPs Treated Wheat Plants

A pot experiment was conducted in a climate-controlled environment to evaluate the physiological parameters of wheat. Each pot contained 200 g of sterilized sand and received 25 mL of half-strength Hoagland solution [64]. Seeds of the FSD 2008 wheat variety were sterilized by soaking in 100 mL of 0.1 N bleach solution for 15 min, followed by four 10-min rinses with sterile dH_2_O. All seed treatments were performed as previously described. The seeds were air-dried under laminar flow, placed on 1% water-agar plates, and incubated at 22–24 °C. After three days, the germinated seedlings were transferred individually to the pots. The plants were maintained in a climate-controlled room with 60% relative humidity and a 09-h photoperiod (200 µM·m^−2^·s^−1^ at pot height using fluorescent lights, 15/20 °C). The experiment used a completely randomized block design (CRBD). Room conditions were set at 20 ± 2 °C, with a light source of 6000 ± 500 FLUX for 09 ± 1 h daily. Plants were watered every two days with autoclaved distilled water, and a second application of half-strength Hoagland’s solution was administered after 15 days to ensure adequate moisture and nutrients. All plants were harvested after 40 days and assessed for total protein, carbohydrate, phenol, and chlorophyll contents. Total chlorophyll content was estimated using Aron’s classical method with a mixture of acetone and water, as described by Manolopoulou et al. [65]. To estimate total protein content, 100 mg of wheat leaves were ground in 50 mM potassium phosphate buffer. The homogenate was centrifuged at 4480× *g* for 20 min, and protein content was determined using Lowry’s method, as described by Mughal et al. [66]. Total soluble sugars in plants were determined using a modified method proposed by Teferea et al. [67]. Total phenolics were estimated in a reaction mixture containing 250 μL of Folin–Ciocalteu reagent, 3 mL of deionized H_2_O, 0.75 mL of 20% Na₂CO₃, and 100 µL of the leaf extract, and calculated using gallic acid as a standard [64]. The 1,1-Diphenyl-2-picrylhydrazyl (DPPH) radical-scavenging capacity of wheat extracts was calculated using a colorimetric reduction assay [68]. The radical scavenging activity was calculated as follows:(4)DPPH Scavenging (%) = (abs t = 0 − abs t = 30)/ abs t = 0 × 100
where: abs (*t* = 0) = (DPPH solution without antioxidant or standard) at *t* = 0 min, abs (*t* = 30) = (absorbance of DPPH + phenolic extracts or samples) at *t* = 30 min.

### 4.12. Statistical Analysis

Experiments were conducted in triplicate to ensure the uniformity and authenticity of the results. The datasets were grouped, and ANOVA was performed using IBM SPSS Statistics Version 23.0.

## 5. Conclusions

This study showcased the effective synthesis and implementation of SR9AgNPs, an innovative silver-based nanofertilizer synthesized using an auxin-producing *Enterococcus* sp. SR9 strain. When SR9 and SR9AgNPs were used together, they markedly improved seed germination, vigor, and biomass in wheat, cucumber, and tomato plants, outperforming both untreated controls and individual treatments. A concentration of 100 ppm SR9AgNPs was safe and highly effective in boosting plant growth and physiological functions, and it included elevated levels of chlorophyll, sugars, and phenolic compounds. The collaborative effects of *Enterococcus* sp. SR9 and nanoparticles are promising eco-friendly approaches for enhancing crop yields. Additional studies are required to assess its performance under field conditions and potential ecological toxicities and to develop commercial nano-biofertilizers for widespread use in agriculture.

## Figures and Tables

**Figure 1 plants-13-02875-f001:**
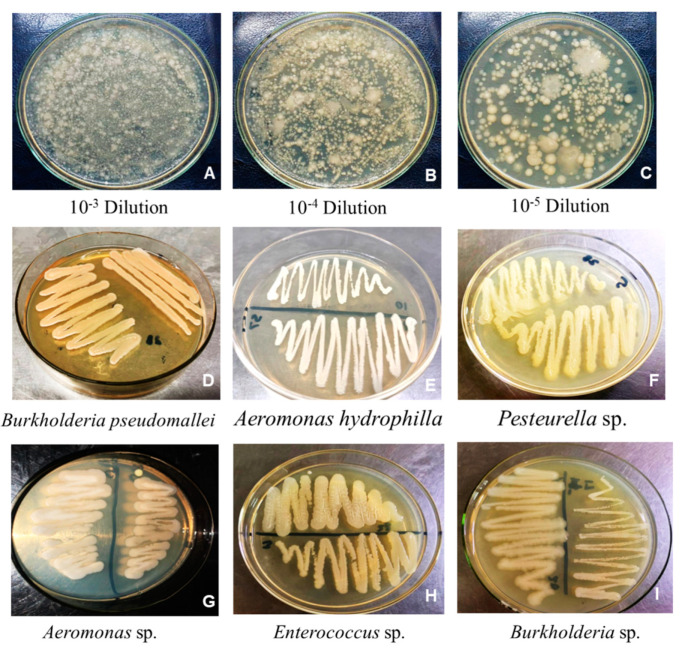
Isolation of the spinach rhizobacteria using a serial dilution method (**A**–**C**). Purified bacterial isolates (**D**–**I**).

**Figure 2 plants-13-02875-f002:**
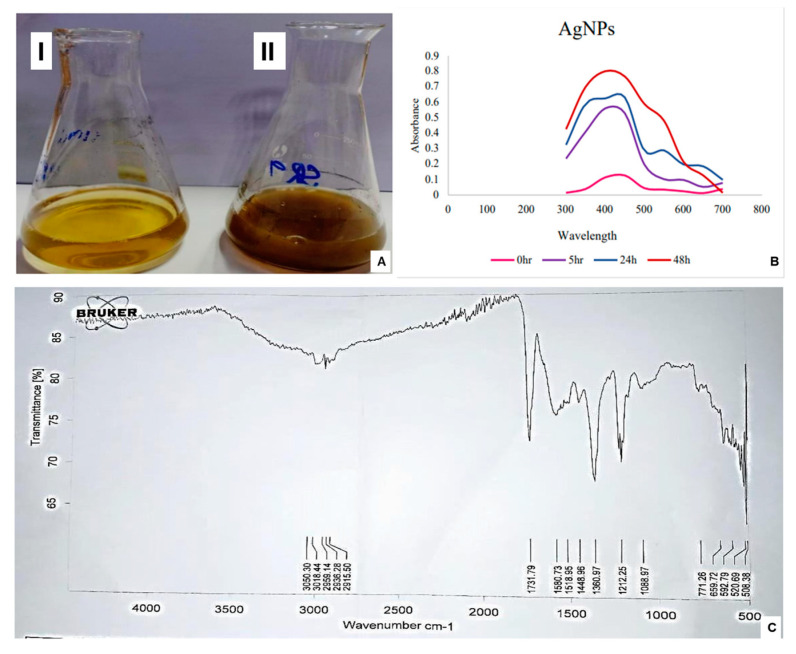
Synthesis of SR9AgNPs (**A**) reaction mixture after 0 h (I), reaction mixture after 48 h (II), UV-VIS Spectrophotometric Analysis of SR9AgNPs (**B**), and FTIR analysis of SR9AgNPs (**C**). The figure illustrates the representative images of the triplicate experiments.

**Figure 3 plants-13-02875-f003:**
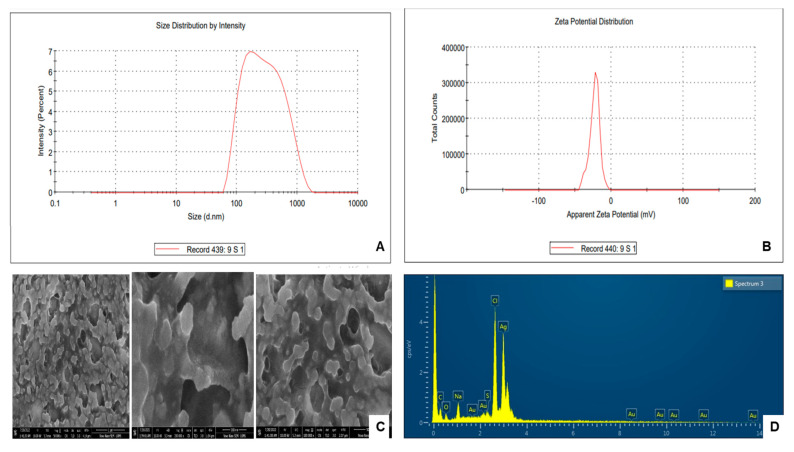
Zeta size pattern of SR9AgNPs (**A**), and Zeta potential distribution of SR9AgNPs (**B**). Scanning Electron Micrograph Image (**C**) and EDX-spectra (**D**) of SR9AgNPs showing the presence of different elements. The figure illustrates the representative images of the triplicate experiments.

**Figure 4 plants-13-02875-f004:**
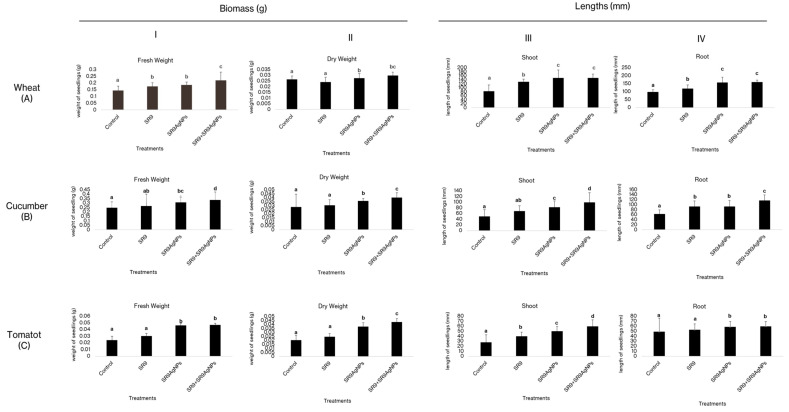
Effect of SR9AgNPs on the wheat ((**A**) I–IV), cucumber ((**B**) I–IV), and tomato ((**C**) I–IV) seedlings after 14 days of incubation. Different letters on the graph bars indicate the significant difference among groups. Similar letters indicate non-significant differences. The data represent average values (*n* = 30 biological replicates). Error bars indicate the standard variation. Tukey’s Test and ANOVA were used to calculate the statistical significance of datasets using SPSS version 23.0.

**Figure 5 plants-13-02875-f005:**
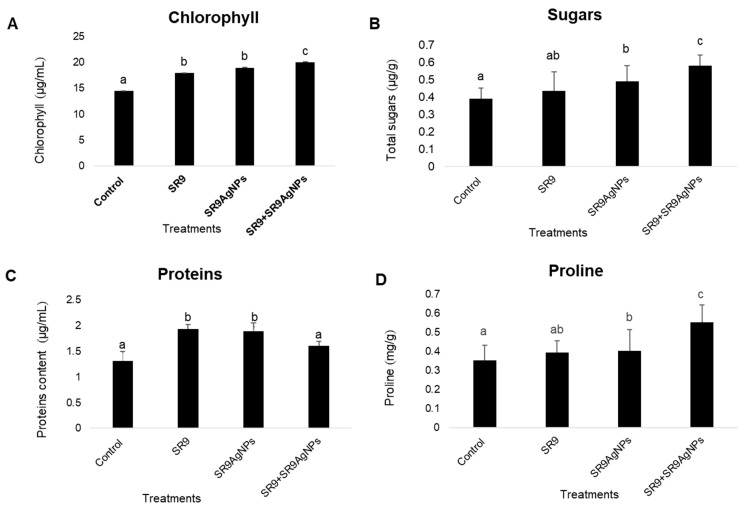
Estimation of physiological parameters of SR9AgNPs treated and untreated wheat plants. The data were collected from 30-day-old plants and indicated the total chlorophyll content (**A**), total soluble sugars (**B**), proteins (**C**), and phenolic content (**D**) of the plants. Different letters on the graph bars indicate the significant difference among groups. Similar letters indicate non-significant differences. The data represent the average values (*n* = 30 biological replicates). Error bars indicate the standard variation. Tukey´s Test and ANOVA were used to calculate the statistical significance of datasets using SPSS version 23.0.

**Figure 6 plants-13-02875-f006:**
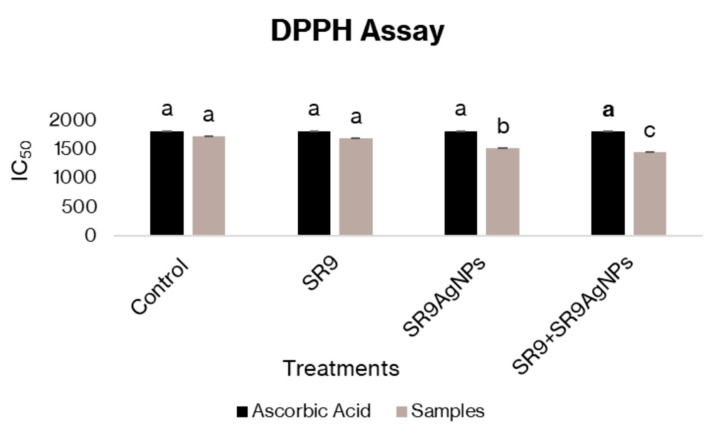
Estimation of the antioxidant potential of SR9AgNPs-treated and untreated wheat plants. The graph shows the measurement of DPPH (2,2-diphenyl-1-picrylhydrazyl) radical scavenging potential. Black bars represent the control of ascorbic acid control, whereas the grey bars are the treatments. The lowest absorbance values in SR9 + SR9AgNPs treated plants indicate the highest antioxidant activity of the treatment used. Different letters on the graph bars indicate the significant difference among groups. Tukey´s Test and ANOVA were used to calculate the statistical significance of datasets using SPSS version 23.0.

**Table 1 plants-13-02875-t001:** Plant growth-promoting parameters of spinach rhizobacteria.

Sr. No.	Bacterial Isolates	Hydrolytic Enzymes					Mineral Solubilization						IAA Production
		Lipase	Amylase	Protease	Catalase	Oxidase	Zinc		Potassium		Phosphate		
							ZnSO_4_	ZnCO_3_	K_2_SO_4_	KH_2_PO_4_	PVK ^1^	NBRIP ^2^	
**1**	SR1	+++	++	+++	+	+	-	+	++	++	++	++	-
**2**	SR2	+++	++	+++	+	+	+	-	++	++	+	++	++
**3**	SR3	+++	+++	++	+	+	-	++	++	++	-	++	+++
**4**	SR4	-	-	-	+	-	-	-	-	-	-	-	-
**5**	SR5	++	-	++	-	-	-	-	-	++	+	+	++
**6**	SR6	-	-	-	+	+	-	-	++	++	+	+	+
**7**	SR7	-	++	-	-	+	+	++	++	++	-	-	+
**8**	SR8	-	++	-	+	-	+	++	++	+	+	-	+
**9**	SR9	+++	+++	+++	+	+	+	+++	++	++	++	++	+++
**10**	SR10	+++	+++	+++	+	+	+	+++	++	++	+	++	+++
**11**	SR11	+++	+++	+++	+	+	-	-	++	++	+	++	+
**12**	SR12	-	-	-	-	+	-	++	++	++	-	++	-

+, positive; -, negative; ++, high positive; +++, highest positive, ^1^ PVK, Pikovskya’s agar, ^2^ NBRIP = National Botanical Research Institute’s Phosphate Growth Medium.

**Table 2 plants-13-02875-t002:** Cytotoxicity assessment of *Enterococcus* sp. SR9AgNPs on germination and vigor index of wheat seedlings.

Concentration of SR9AgNPs (ppm)	Germination (%)	Vigor Index
50	96 ^a^	6790 ± 671.06 ^a^
100	98 ^a^	8823 ± 850.95 ^b^
200	91 ^a^	5200 ± 336.85 ^c^
500	90 ^a^	4580 ± 473.63 ^c^
1000	79 ^b^	3500 ± 413.03 ^d^

Mean (± Standard Deviation) values of germination percentage of seeds treated with different concentrations of SR9AgNPs. Means followed by different letters differed significantly according to the Least Significant Difference (LSD) test (*p* < 0.05).

**Table 3 plants-13-02875-t003:** Germination percentage and vigor indices of plants treated with biofertilizer, nanofertilizer, and a combination of the two.

Crops	Parameters	Control	SR9	SR9AgNPs	SR9 + SR9AgNPs
Wheat	Germination %	90 ^a^	97 ^b^	99 ^b^	99.9 ^b^
	Vigor Index	16,300 ± 774.11 ^a^	25,000 ± 663.04 ^b^	29,820 ± 504.9 ^b^	30,300 ± 309.11 ^c^
Cucumber	Germination %	70 ^a^	86.7 ^b^	90 ^b^	100 ^c^
	Vigor Index	10,240 ± 331.01 ^a^	13,111 ± 514.09 ^a^	17,130 ± 611.01 ^c^	18,640 ± 504.04 ^c^
Tomato	Germination %	80 ^a^	86 ^a^	90 ^b^	100 ^c^
	Vigor Index	6110 ± 313.16 ^a^	7637 ± 375.01 ^b^	8720 ± 374.07 ^c^	9890 ± 463.31 ^d^

Mean (± Standard Deviation) values of the germination percentage of seeds used for plant growth analysis. Means followed by different letters differed significantly according to the Least Significant Difference (LSD) test (*p* < 0.05).

## Data Availability

All associated data are provided within the article.

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
