# Peer review of "A Novel Enterococcus-Based Nanofertilizer Promotes Seedling Growth and Vigor in Wheat (Triticum aestivum L.)"

_plants, 2024, doi:10.3390/plants13202875_

Round 1
Reviewer 1 Report
Comments and Suggestions for Authors
Dear authors, i read with interest the article. In my opinion the article can be improved especially in the introduction, M&M, and conclusion section. Discussion are very good.
I suggest a major revision

Author Response
Comment 1- Line 39: Please add a reference.
Response: References 1 and 2 are added as suggested by the reviewer.
Comment 2- Please explain better the term bio fertilezer and add some ref, explore the different type of novel biofertilezer. some explample are here:
https://doi.org/10.1007/s10668-023-03117-z, https://doi.org/10.3390/plants13101335
Response: Authors have added the said explanation of biofertilizers and have added the relevant citations. The explanation can be found from lines 46-49 and the reference numbers are 6 and 7.
Comment 3- Lines 66-68: Explain this concept and add some references.
Response: The explanation of the sentence is added from the lines 72-76 and the references added are numbered from 17-19.
Comment 4- Kumar and collaborators change to Kumar et al.
Response: Kumar and collaborators change to Kumar et al. in line 88.
Comment 5- Figure 4: Figures are not readable, please make it bigger.
Response: Figure legends are revised for figure 4 and the figure size is also enlarged.
Comment 6- Figure 6: Treatments?
Response: Series name has been changed to Samples from Treatments as advised by the reviewer.
Comment 7- Discussion is very short, please go deeper with other studies.
Response: Discussion has been revised and all the changes with the new citations are highlighted in yellow to be reviewed.
Comment 8- Spinach (Spinacia oleracea)
Response: The sentence is rephrased as asked by the reviewer in Line 323.
Comment 9- Use formula style for the equation.
Response: The equation 1 in Line 378 has been changed to formula style.
Comment 10- please add a ref for germination rate like this https://doi.org/10.3390/app14020631.
Response: The reference in Line 402 for Germination index has been changed with the one proposed by the reviewer and is reference number 56.
Comment 11- Heading 4.11: Be more specific about the methods used for the calculation of this parameters.
Response: The mentioned heading has been rephrased and all the methods used have the relevant citations.
Comment 12- Use formula style for the equation.
Response: The equation 4 in Line 500 has been changed to formula style.
Comment 13- Conclusion: Please add some of the main results in the conclusion.
Response: Conclusion section is revised.
Reviewer 2 Report
Comments and Suggestions for Authors
Plants-3172191
Evaluation Report COMMENTS REFEREE X
Notation
ave average
CVar coefficient of variation
LCA life cycle assessment
LL line or lines
MS manuscript
NP nanoparticles
PEI potential environmental impact
PGPB plant growth promoting bacteria
PP page or pages
R/D research and development
RMS revised manuscript
s standard deviation
Greek characters
a Type I error in a statistical test, typically 0.01 or 0.05
b Type II error in a statistical test, typically lower than 0.15
Preface
The numbers of the lines or pages referred to in my Comments belong to the attached Word file Annotated MS. Probably the line numbers do not necessarily coincide with the numbers in the original pdf file prepared by MDPI. Hopefully they will be congruent, but I doubt it.
Please be aware that the main evaluation report is complemented by three annexes at the end of the document: Annex 1. English revision of Abstract with Trinka; Annex 2. Revision of selected statistical calculations; and Annex 3. Annotated Manuscript
Content
Notation 1
Preface 1
1.Evaluation report 3
2.Annex 1. English revision of Abstract with Trinka 17
3.Annex 2. Revision of selected statistical calculations 20
4.Annex 3. Annotated Manuscript 22
Evaluation Report
1.Overall judgement and recommendations
This is a very interesting article on a subject sustantive of a “hot” topics such as nanoagriculture. A good thermometer of the increasing interest in nanoagriculture is the flurry of Reviews in the last 5 years (2018-2023)
However, I have to recommend Major Revision. The revised manuscript RMS should Include the following (plus other features in the particular Comments and the Annotated MS):
1.1. Make an effort to eliminate several overstatements in the MS
1.2. Statistical calculations are of concern, particularly the standard deviations of the variable Vigor. Referee X has made his/her own calculations and even with low coefficients of variations of 1% in the variables Germination and Plant length is impossible to achieve such low standard deviations as reported in the MS.
Authors are kindly asked to report detailed raw data and detailed statistical calculations in experiments with Germination index, Plant length, and Vigor in the Supplementary Material of the RMS. Reviewer will check them for ruling out any systematic or accidental error of calculation. If any error is detected, Authors will be asked to change numbers in the text and tables of the MS and re-interpretation of results, Authors are kindly asked to conduct a thorough and consistent revision in the RMS.
1.3. The Error type I (alpha) and Error type II (beta) should be reported for each statistical test.
1.4. The MS is permeated with an overstatement trend. There are passages of overstatement in the Introduction, Results, Discussion, and Conclusion. This trend should be eliminated in the RMS. Science is not the place for unintended sales attitude or pompous style.
1.5. Discussion is short and superficial.
1.5.1. Efforts should be made to discuss results based on biological, environmental, and toxicity arguments that could lead, in time, to new hypotheses and to the advancement of the R/D. Some results (for instance, the significant interaction of the combined treatment SR9-SR9AgNP that gives the best results for several variables representative of seed emergence and plant growth) are referred to one sentence stating as a generic “synergism.” This is an example where more elaboration is needed.
1.5.2. The Discussion should be enriched with information and critical
interpretation on the cost, environmental impacts, human health implications. and circularity of the current technology proposed in the MS. So far, the Authors have superficially mentioned a few References on the cost and presumably low plant toxicity of other cases, most of them from other systems of microbial-heavy metal NPs.
There is no information on these issues for the particular research of the MS. Please provide them in the RMS to the best of your efforts and knowledge.
1.6. Conclusion is short and uninformative. At least, Conclusion should have an extension similar to Abstract or larger. Also, it must cover in more detail the results on the experiments reported in the Results section.
The Conclusion should be re-written.
1.7. Methodology: some experiments are vague and do not comply with requisites for reproducibility of the experiments that is crucial in scientific method. For instance, the experiment in sub-section 4.11 should be re-written. Other experiments should be completed with more detailed statistical information on the statistical tests performed, for example, and other info.
So far, the current Methodology does not comply with the requirement of reproducibility mandated by the scientific method.
1.8. Moderate revision of English required (submit report of Grammarly or Trinka, etc. to the Reviewers) with a score 98% or higher. Alternatively, please provide a Certificate from a recognized provider of scientific English translation and style revision services. More comments on this below.
1.9. Proof of similarity check required (submit report of Ithenticate, Turnitin, similar) Authors should submit to the Reviewers evidence of Similarities check lower than 7%. More comments on this below.
1.10. Several references that are key to the Methodology lack their corresponding DOI. Please provide the DOI.
OTHERWISE IT WILL CONSIDERED A GREY REFERENCE AND THE AUTHORS ARE KINDLY ASKED EITHER TO INCLUDE AN ALTERNATATIVE OPEN LITERATURE REFERENCE OR TO WRITE THE COMPLETE TECHNIQUE IN THE SUPPLEMENTARY MATERIAL DOCUMENT
1.11. Please number all the equations consecutively, with number between brackets. The la latter can be curved or square. I prefer square brackets.
1.12. Several tables miss the footnotes and the standard deviation of average results. Please complete them.
1.13. Please re-submit revised manuscript following my remarks. If you do not agree with any particular Comment of mine, please defend accordingly with References and sound arguments.
In spite to one mention to Sustainability at the start of the article in the justification of the rationale, in this paper there are no reports on available information on the sustainability analysis of nanoagriculture and nanobiofertilizers in the framework of PEI and LCA or on the basis of another quantitative system analysis technique. In fact, Sustainability is out of the scope of the present version of the MS. So, the possible relationship between this article and Sustainability is only a matter of opinion.
Consequently, I strongly recommend to delete the word sustainability from this paper. My recommendation to Authors for the future who wish to support their research with sustainability quotation, is that Sustainability should be demonstrated with facts and numbers. Sustainability is not an opinion, it is not a fashion, it is not a mere wish, it is not an ornament.
After these modifications, the Abstract and Conclusion should be rewritten.
2. Notation table
Please include a Table named Notation where all the abbreviations, acronyms, Greek characters, and symbols used throughout the MS are gathered in alphabetical order, with their corresponding meanings. Place Notation table before the List of References.
Please note that this MS contains nearly 50 abbreviations or more (only with the biochemical parameters there are 15 or more abbreviations) and it is not possible for a reader to memorize all the meanings or the place where the abbreviation was spelled it out in the text of the MS for the first time. Therefore, a central Notation table is needed to keep the abbreviations in a known place.
The Notation table can contain a subsection labelled Greek characters where the Greek symbols and their meanings in the MS are included, in alphabetical order. Please check some good Dictionary for the order of letters in the Greek alphabet.
Notation table will help Editors, Reviewers, and readership to rapidly locate an abbreviation or a symbol and learn its meaning. It facilitates very much the easy reading of any MS and the task of Reviewers.
Whether the Notation table is published or not, it is a matter of the Journal policy. I firmly recommend the publication of the Notation table.
3. Similarity and self similarity check. The Authors should send evidence of an analysis of similarity and self similarity of the MS. Please use Ithenticate or another recognized software for checking similarities in the MS.
I feel that in the 21st century, it is a mandatory task of the Authors to show that similarity and self similarity are low when submitting the MS. I recommend a proportion of 7% or lower.
Prepare the file for the check eliminating
figures and tables (only leave the titles of tables and legends of figures),
the Names of the authors,
their affiliations,
the keywords,
the Notation table,
the logos of MDPI and MDPI formatting that are repeated on each page,
the Acknowldegements section
the sections on several Declarations such as Conflict of interest, Funding, etc., and
the list of References.
Configuration settings in Ithenticate:
Tick the box that asks for neglecting the similarities of 9 words in a row or less
tick the box that requests neglecting the citations of references in the text.
These categories can account for a considerable portion of the similarities and it they are not removed is unfair to the article.
4. English usage
Please be aware that Referee X is a non native English speaker. Therefore, his (her) capabilities to evaluate English usage in the MS is moderate.
I recommended Moderate Revision of English. Since I have free Grammarly integrated to my Word in the computer. I could see that Grammarly automatically detected many mistakes in the MS (the light ones but the most frequent: singular-plural mismatches, preposition use, conjunction use, use of commas and semicolon, order of complex adjective-noun “the process of type 1” better write “the type 1 process”, etc.) Free Grammarly cannot detect/correct awkward syntax. Anyway, I could detect some sentences with awkward syntax, probably some “Spanishisms”. I did not correct them. Let Grammarly do its work.
As a sample and aid, I included in Annex 1 at the end of this Evaluation copies of the Abstract before and after English revision with Trinka.
Please ask the Authors to provide evidence of English usage revision. Either a certificate of English revision by a recognized provider of English services or the Report of Grammarly Premium or Professional revision with a 98% or higher score will be adequate. If the Authors choose the software, typically two rounds of revision in Grammarly would achieve that score. I recommend two rounds of revision by Grammarly and one round of revision by Trinka.
In this way, English usage evaluation will be in the hands of experts.
5. An Index of the Review with Dewey numbering is highly recommended. It can be placed between the keywords and the Introduction of the paper. The index will help the readership to determine the scope of the article at a glance and Reviewers to rapidly locate the place of a given section or sub-section. It will also help the Authors to keep the order in their RMS.
Finally, my recommendation to Editors is to publish both the Notation table and the Index.
Here-in after, please consult the Annotated Manuscript, Annex 3.
6. Title and keywords: Too generic, Readers can believe that the article is valid for all the plants. Please add “…wheat and tomatos” to the title and keywords. Please see my annotation in the annotated MS
7. Abstract.
7.1. It reads “Toxicity assessment of SR9AgNPs demonstrated their biosafety at a concentration of 100 ppm…” This is the first example of overstatement. Only some simple toxicity tests were performed dealing with the effect of SR9AgNP concentration on plant germination at lab scale. Biosafety is much more than that. Please change to
“…Toxicity assessment of SR9AgNPs demonstrated no apparent toxicity to germination of wheat seeds at a concentration of 100 ppm….”
7,2, Please correct Abstract for English usage (as well as the whole MS).
Particularly, the English of the Abstract should be impeccable because it is the first impression that the reader gets re the article. An there is only one opportunity to give a good first impression.
As a sample and aid, I included in Annex 1 at the end of this Evaluation report the copies of the Abstract before and after English revision with Trinka. Authors are kindly asked to revise the English of the Abstract (as well the complete body of the article) with either Grammarly Premimum/Professional or a recognized company that provides English editing services.
Furthermore, if the RMS is modified as recommended, then there could be changes in results and discussion (for instance, after revision of statistical calculations) and the Abstract should be rewritten.
8.Introduction
8.1. LL36 Eliminate the word Sustainability. Do not use concepts that your article does not support. The article does not show that the complex Silver-bacteria is sustainable or contributes to the sustainability of agriculture. Please eliminate the word “sustainability” and rephrase. Please read my comments on Sustainability on the page 4 at the bottom. This is an example of overstatement
8.2. The goal of the current research was to determine the performance at lab scale of a naobioferilizer based on silver. This is very debatable and counter-intuitive from the environmental point of view. Please elaborate. Heavy metals should be removed from the environment because of their toxicity, and the problem is segregating and sequestering them (high costs and high environmental impacts of these processes as well).
Spreading nanoparticles of heavy metals is quite contrary to the removal requirements (it is just the opposite). It is actually counter intuitive from the process and environmental points of view, and could lead to spreading hazards to health and environment.
It is also contrary to environmental sustainability. The Authors should discuss this issue as part of the rationale (the other, dark face of their rationale)..
8.3. LL 54-44 the abbreviation does not correspond to PGPR, it should read PGPB. Please correct and place the abbreviation in the Notation table. Please also correct the abbreviation throughout the remainder of the RMS
8.4. LL 58-60. Please rephrase, consider “…Nano-biofertilizers consist of delivery agents and specific compounds that are synthesized with the help of PGPB microbes….”
8.5. LL 62. Please rephrase; ‘These’ is ambiguous. Consider “Biofertilizers can be designed using bottom-up and top-down approaches…”
8.6. LL 64-65. This statement is crucial for the status and future of nanobiofertilizers. Please provide two of three more references from the open literature that supports the ideas of the statement.
8.7. LL 66. This part of the statement is also crucial for the status and future development of nanobiofertilizers. Please provide two more references from the open literature that supports the idea of this fragment.
8.8. LL 72-73. Please rephrase. Consider “The uptake of NP by plants takes place by binding….”
8.9.LL 79-81. Please replace the word ‘non-advantageous’ by the more common ‘deleterious’ or ‘negative’, especially when referring to effects.
8.10. LL 88-89. Please consider replacing ‘modifying’ by ‘increasing’.
8.11. LL 91-93. Please more elaboration on negative issues regarding silver, silver nitrate, the microbial-mediated AgNP fabrication process, etc., are required.
What about the costs? Silver is not cheap. Silver nitrate is not cheap. and options to have soluble Ag+ cations are restricted because other silver salts such as sodium chloride and sodium sulphate are insoluble.
What about the effectiveness of using commercial grade silver nitrate solutions ? This could abate the costs, but there could be negative effects on the amount of synthesized NP and other effects of impurities?
Any reference or idea about the relative costs of chemical grade and commercial grade silver nitrate?
And a crucial issue, what about the potential environmental impacts on silver handling and nanoparticle fabrication, including human health effects of the overall chain of silver production (mining, melting, purification) plus the impacts of the microbially-mediated fabrication of AgNP itself?
Elaborate briefly on the needs of determining the potential environmental impacts (PEIs) in a holistic approach, from the cradle to grave. Silver mining (as a part of the process) is known to be very polluting with several adverse environmental and human health impacts. Not only the silver toxicity is of concern. Also, the cyanide used in the silver purification stage is of environmental concern, as an example. Alternatively, if this elaboration becomes long, place it in Supplementary Material document as an Annex, previous citation of the Annex in the text of the RMS. The toxicity of the AgNP per se is not the only source of PEIs in the life cycle of AgNPs fabrication and use.
9. Methodology Section 2 new number
9.1. This section is placed after sections ‘Results’and ‘Discussion’ in the original MS. The standard order of journal Fermentation is ‘Methodology’ first. I recommend to change the order of the sections. Also, please change the order and number of the other Sections as follows:
Section 2. Methodology, Section 3. Results, Section 4. Discussion, Section 5, Conclusion throughout the RMS
9.2. LL 423-424 Section 4.8. SR9AgNP: doubts about the nature of the NPs.
9.2.1.Do you use the whole mixture of cells and bacteria-mediated synthesized NPs?
9.2.2. Do you use only the separated bacteria-mediated synthesized NPs?
9.2.3. Where are the SR9AgNP, in the supernatant or in the pellet?
9.2.4. If the SR9AgNP are in the supernatant, How was determined that the centrifugation and filtration of supernatant does remove only the cells of the micoorganism but does not remove the silver NP? References? Previous tests, unpublished?
9.2.5.How do the authors know whether there are no silver NPs attached to the surface of the bacterial cells (nanodecorated bacteria)?
PLEASE EXPLAIN AND ADDRESS ALL THESE QUESTIONS SUCCINTLY HERE OR IN SUPPLEMENTARY MATERIAL. To clarify all these issues is crucial to the reproducibility of the experiment, as mandated by scientific method. I feel that the information on this experiment is sketchy and cannot warrant the reproduction or repetition of the experiment.
9.3. Please number all the equations in consecutive order. Most equations appear in Methodology
9.4 Regarding the propagation of SR9 and the SR9-assisted synthesis of AgNP tests, where they conducted aerobically or anoxically? Please state the type of culturing in the corresponding sub-sections of the Methology.
9.5. LL 379 to 382. I could not find the selection procedure description of isolates in the Methdology or in the Results sectIons of the MS. Please describe the selection procedure and criteria in the Methodology section. It is mandatory for reproducibility-sake.
9.6. LL 388. Please homogenize the use of the abbreviation PGPB. Sometimes PGPP or PGPR are used. I proposed PGPB because the expression is plant growth promoting bacteria.
9.7. LL 475-493 Section 4.11. It is unclear if this test was carried out in pots, the size of the pots, irrigation rate, type of water, illumination regime, type of soil or hydroponics, etc. Or the experiment was performed in Petri dishes?
Please give full info, sub-section 4.11 should be re-written.
9.8. LL 483 Missing the number and in the list of referenes of the reference by Mughal et al. Please number the reference and include the reference in the List of References.
9.9. LL 487-492. More explanations regarding the terms in the equation for DPPH Scavenging. Number the equation. Please explain the meaning of radical + methanol, and radical + phenolic extracts and the corresponding techniques/preparation procedure to get them; I assume that they mean radical solution and methanol extracts, but I am not sure.
Moreover, I assume that the DPPH Scavenging technique is in the Reference 57. However, there is no DOI of this Reference.
Please provide a DOI, OTHERWISE IT WILL CONSIDERED A GREY REFERENCE AND THE AUTHORS ARE KINDLY ASKED EITHER TO INCLUDE AN ALTERNATATIVE OPEN LITERATURE REFERENCE OR TO WRITE THE COMPLETE TECHNIQUE IN THE SUPPLEMENTARY MATERIAL DOCUMENT
9.10. Please report Error type I and Error type II probabilities for ALL statistical experiments and comparisons described in the Methdology.
10 . Section 3 Results New number
10.1. Title LL100 Please change the place and the number (use number 3) of the Secion Results. Change the numbers of sub-sections accordingly, i.e., 3.1., 3.2., etc.
10.2. LL109 Subsection 3.1. Please clarify how there are 12 isolates but the Fig. 1 only has 6 strains.
10.3. Table 1 Subsection 3.1.; please consider modifying the footnotes, see annotated article Annex 3.
10.4. LL136 & ff Please conclude this sub-section with the pre-selection of the five bacterial candidates and how the pre-selection was performed;
What was the criterion used? higher percentage of parameters in table 1? Weighed higher percentage of parameters in table 1?
Please write the procedure or pre-selection in a detailed form and the corresponding equations used, if any, in Mehtodology (your section 4 now section 2)
10.5. LL158 & ff Subsection 3.3. Please report the proportion of conversion of silver cation available in the silver nitrate solution to silver in the NP, very important, in percent.
10.6. LL161 Subsection 3.3. “…a 2 mM AgNO3…”, please add between parentheses the concentration in mg/L, that is “…a 2 mM AgNO3 (340 mg/L….” because Readers from the environmental science field can be more comfortable with the units.
10.7. LL169 Subsection 3.3.
10.7.1. PDI, please spell it out on the left of the abbreviations. Please include the abbreviation in Notation.
10.7.2. What are the units of 0.2? Or is it the coefficient of variation, dimensionless?
10.8. LL171-173 Sub-section 3.4. It reads “SR9AgNPs Can be Successfully Applied as a Nanofertilizer”
PLEASE REPHRASE, it is misleading and Readers could believe that the nanofertilizing feature of the bact-AgNP was demonstrated. Since the test was at lab scale ONLY THE POTENTIAL OF BACT-AgNP AS NANOFERTILIZER AT LAB SCALE WAS SHOWN HERE. There is an uncomfortable trend of this paper to overstate the meaning of results.
Examples: the non-supported use of Sustainability feature, using the concept biosafety when only the cytotoxic effects were determined, now the overstated title of sub-section 3.4. The Authors should be more careful to control this issue since it detracts from the quality of the paper. Science should be separated from unintended sales attitude, or pompous style.
10.10. LL.203 & ff, Table 2, Sub-section 3.4.
We think that this part is an issue of major concern re results.
10.10.1.Please provide standard deviations of Germination Index
10.10.2.Missing data of plant length, averages and standard deviations, please provide averages and std deviations in two more columns of the Table.
10.10.4. Missing the footnotes. Please complete.
10.10.5.Please report the units of plant length, presumably in mm (?)
Interestingly the units of Vigor are (%)*mm or (%)*cm, where (%) comes from the germination percentage and the unit of length comes form the plant length, either in mm or cm.
10.10.6. PLEASE VERIFY CALCULATIONS OF THE STANDARD DEVIATIONS OF VIGOR. Low standard deviations of Vigor Index. How? Vigor Index is calculated as the product of two magnitudes. Each magnitude has an error (std dev). The final standard deviation of the product typically is higher than the standard deviations of the factors, please see Kreyszig, E. (1970). Introductory Mathematical Statistics, Chapter 19 Measurement Errors, John Wiley & Sons, Inc. New York, USA. Or another book on Statistics that includes Analysis of Errors and Propagation of Errors.
The standard deviation of Vigor can be estimated with Eq. RevX1 below (Kreyszig, 1970; Chapter 19)
Sproduct = SQRT (f2^2*Sf1^2 + f1^2 Sf2^2) [RevX 1]
where f1 is factor 1, f2 is factor 2, S are the std dev of the product and factors (subindices), SQRT is the square root of the expression between parentheses.
A simplified rule for the coefficient of variation of a product when the coeffs. of variation of the factors are equal, is
Coeff. var.product= 1.4142*Coeff. var.factor = either 7.07 or 4.24% [RefX 2]
The above results correspond to factor C Var of 5 and 3%, respectively, and they are
higher than 5% and 3%, respectively.
Please recall that
C Var (%) = s/ave*100 [RefX 3]
where s is the standard deviation, and ave is the average of data
My own calculations predicted standard deviations of Vigor in the order of E02, from 250 to 620 when the coefficients of variation of both germination index and plant length are 5%, in the order of 250 to 370 when both factors have C Var 3%, and in the order of 50 to 125 when both factors have coefficients of variation 1%. Please see Excel file in Annex 2 at the end of this Evaluation report.
This is a confirmation that in general, both the Coeff. of var. and the std. dev. of a product are higher than those of the individual factors. Colloquially, propagation of errors of a magnitude that is calculated by algebraic operations of magnitudes with error follows the principle of the “snow ball descending on a hill”.
Reductio ad absurdum: Here we proceed backwards with the mathematical demonstration. We depart from Authors’ result of the standard deviation and we will obtain an extremely low C Var (absurd) for any experimental factor.
The std dev of vigor shown in Table 1 for the best treatment is 0.03 according to Authors.
This means a Coef. Of var. of vigor 0.03/8820*100 = 0.00034%. Assuming that both the Germination and plant length have the same Coef. of var., then we would have coeff. of var of either the Germination factor or the Plant length (factors) equal to 0.00024%, unbelievable precise!!!!!!
10.10.7. There is a noticeable difference between the Vigor of wheat in Table 2 and that in Table 3 (8820 and 29820, respectively). It represents a factor of 3. Please discuss.
Authors are kindly asked to revise carefully the statistical analysis of their data.
Moreover, they are kindly asked to submit the raw data and the statistical analysis for review along the RMS. Reviewer X would like to check and rule out any possibility of systematic or accidental error with the calculations.
10.11. LL.227 & ff Table 3. Sub-section 3.4.
10.11.1. Please report the amounts or concentrations of the additives (100 ppm in all cases???, or varying doses???). Create a column “Dose” in Table 3 to write the concentration of each agent.
10.11.2. Without knowing the doses, comparisons are meaningless. In sub-section 2.10 new number of Methodology (old 4.10) this info is missing, except for the treatment SR9AgNP. The doses used should be updated also in Methodology.
10.11.3. Again, the standard deviations of Vigor are extremely low; the coefficients of variation are in the order of thousandths of percent (!!!!!)
With these low values of CVar and std. dev. of vigor, the test of means for vigor will give significantly different treatment means for all the means or for most means.
10.11.4. Why the outstanding difference of Vigor in this test (for SR9AgN) and the test in Table 2, i.e., 29820 units of Vigor and 88320 units of Vigor, respectively? Please explain. It is a factor of 3, approximately
10.11.5. In several variables and for selected plant(s), the “mixed” treatment SR9 plus SR9AgNP showed the highest results (and significant). In statistical terms, this is an interaction. What is the interpretation on the possible biological and environmental causes that could explain the statistical interaction? Please explain. The same for results in Table 2 that should be discussed in first principles of biology, environmental science, toxicity.
10.12. LL.231-238 Figure 4.
10.12.1. Please report the standard error of the experiment for each response variable, either in the text or in the legend of the Figures.
I think that the standard errors of the experiments are very low, i.e., the experiments were sufficiently sensitive.
10.12.2. The error bars in Fig. 4 are imperceptible. Please make a statement in the figure legend something like “…the coef. of variation of length were lower than xx, zz, and ww% for shoots of wheat, cucumber, and tomato, respectively…”The C Var of length were lower than pp, qq, and ss% for roots of wheat, cucumber, and tomato, respectively..”
10.12.3. Please increase the size of the labels and axis in the Figures. Use bold black letters for labels and axis titles. It seems that some axis titles are grey or perhaps it is the effect of the large reduction of the image.
10.12.4. The doses of each agent used should be reminded and linked to the doses reported in Table 3, for complete info to the Reader. Doses info can be repeated in the legend of the Figure.
10.12.5. In several variables and for selected plant(s), the “mixed” treatment SR9 plus SR9AgNP showed the highest results (and significant). In statistical terms, this is an interaction. What is the interpretation on the possible biological, environmental and toxicity phenomena and effects that could explain the statistical interaction? Please discuss in the Discussion section..
10.13. LL.255-263 Fig. 5; LL 265-266 Fig. 6. Similar Comments to those in 10.12.
11. Section 4 Discussion new number.
11.1. Discussion is short and superficial. Missing discussion of results based on first principles of biology, environmental science, toxicity. As one example, see my comment in Comment 1.3.
Also, Discussion reveals again the trend to overstate the significance of the experimental results. This trend should be eliminated from a scientific paper.
11.2. LL279-280 it reads, “However, their combined potential as bionanofertilizer has 279 not been addressed to date to our knowledge.
This statement should be also used in the Abstract or the Introduction to reinforce the originality of the contribution.
11.3 . LL 289-291 it reads “ Similar synergistic effects have 289
been previously reported in the literature, where a substantial increase in 290
plant growth was observed for biofertilizer and nanofertilizer treated plants [34]”
This statement is superficial. There is a need to discuss what are the possible biological, environmental, toxicity causes or phenomena that could explain the statistical interaction between the two treatments. The Authors restrict the discussion to label the result as “synergism” and citing one reference. I feel that the result is surprising and deserves a deeper discussion. This is one example that shows that Discussion should be rewritten in several parts.
11.4. LL 296 it reads “…soybean, and maize when subjected to 2000 mg/L nCeO2….”
What is the meaning of the lowercase n? Nanoparticles? If so, please use NP that is better recognizable. Otherwise you should explain the meaning of ‘n’ in the text and place it in Notation table.
11.5. LL 300-302. It reads “Microbial-assisted nanoparticle biosynthesis is generally considered safe and 300 has shown less cytotoxicity and phytotoxicity compared to those synthesized applying 301 other methodologies [38]
More references are needed to support such a general and crucial statement. It is needed to show that the statement is a current of thought, not an isolated thought.The statement wants to be very general, one reference does not support the intended generality.
11.6. LL 302-305 it reads “In our work, we tested a range of concentrations of SR9AgNPs 302 and observed phytotoxicity at the highest doses, while a 100-ppm concentration showed 303 effective seedling growth and increased seedling vigor and weight, with no negative ef- 304 fects. “
I disagree with this statement. The article shows no toxic effects based merely on two phenomenological variables, Germination and Plant length. Results were promising for a dose of 100 ppm of the agent. However, the test is very short of phytotoxicity
No data has been obtained for phytotoxic effects on root development and other variables in table 2. Again, authors tend to overstate the conclusions with respect to results.
Please rephrase and keep the statement related to the true scope of the experiment.
11.7. LL 309-311 it reads “These results suggest that, at a recommended concentration of 100 ppm, 309 SR9AgNPs can be considered non phytotoxic, and can be recommended for the develop- 310 ment of nano-bioformulations in the future.”
Caution, caution, do not generalize based on a simple experiment such as that in table 2. Again, authors tend to overstate the conclusions with respect to results.
Rephrase and keep the statement consistent with the scope the experiment.
Consider, for example “These results suggest that, at a recommended concentration of 100 ppm, 309 SR9AgNPs can be considered non phytotoxic in a preliminary way, and such NPs can be recommended for the develop- 310 ment of nano-bioformulations in the future.”
12. Conclusion
12.1. Too short, please elaborate more. At least, this section should be as long as the Abstract, or more.
Please be aware that Readers who are very busy they actually read the Abstract and the Conclusion of an article before deciding to read the full paper. Therefore, Conclusion should be a bit more informative than the Abstract.
12.2. Conclusion is singular, there is only one conclusion in one paper; The ‘Conclusion’ section can contain several issues, but the plurality of issues does not convert ‘Conclusion’ into plural. The dog’s tail has nearly 15 to 17 vertebras (atrophied, but vertebras in the end) Therefore, the presence of several vertebras in the tail does not transform the tail into tails. One dog, only one tail. One paper, one section Conclusion.
12.3. LL 500 & ff, the following paragraph is not Conclusion, it is Perspectives or Further work.
I do not recommend to include Perspectives or Further work in an experimental paper.
Research Ideas are confidential as long as they were not executed and they are not published.
Research ideas are not public property, they belong to the Group that generates them.
This is one of the reasons why Editors and Reviewers are committed to treat the evaluation of non published articles with the highest confidentiality.
I learned this from several First World researchers. Of course, it is a matter of belief and opinion.
In contrast, Perspectives or Further work are/is mandatory for Review papers.
13. Tables: Some tables missed the footnotes. Please complete.
Some tables lacked standard deviations of the averages. Please complete.
Some tables lacked average results of selected variables. Please complete.
Please see the annotated manuscript Annex 3,
.
14 References
14.1. References 19/57 articles from the last 5 years (2019-2024) , approx. 33%; low proportion.
14.2. Report DOI of articles in the List of References, for each reference. Whenever the DOI does not exist, replace or add another Reference with DOI to support the issue that was supported by the article without DOI. A paper without DOI is considered to be grey literature and I do not recommend to support Methodology or significant statements based on papers without DOI, very difficult to get.
14.3. Self-citation proportion below 15%, OK.
References
Kreyszig, Erwin (1970). Introductory Mathematical Statistics. John Wiley and Sons, Inc., New York, USA. Chapter 19. There is a Spanish translation by Limusa-Wiley of Mexico, México City.
libgen.is is a website to download free technical, science and engineering books that will be useful to Authors. The book by Kreyszig is not included in that database, but several other valuable books on Statistics that contain sections or chapters on Error Propagation can be found and downloaded.

Moderate revision. Mandatory submit either Certificate of a recognized English service provider or Grammarly Premium/Professional score higher than 98% for examination by the Referee X.
Plants-3172191
Evaluation Report COMMENTS REFEREE X
Notation
ave average
CVar coefficient of variation
LCA life cycle assessment
LL line or lines
MS manuscript
NP nanoparticles
PEI potential environmental impact
PGPB plant growth promoting bacteria
PP page or pages
R/D research and development
RMS revised manuscript
s standard deviation
Greek characters
a Type I error in a statistical test, typically 0.01 or 0.05
b Type II error in a statistical test, typically lower than 0.15
Preface
The numbers of the lines or pages referred to in my Comments belong to the attached Word file Annotated MS. Probably the line numbers do not necessarily coincide with the numbers in the original pdf file prepared by MDPI. Hopefully they will be congruent, but I doubt it.
Please be aware that the main evaluation report is complemented by three annexes at the end of the document: Annex 1. English revision of Abstract with Trinka; Annex 2. Revision of selected statistical calculations; and Annex 3. Annotated Manuscript
Content
Notation 1
Preface 1
1.Evaluation report 3
2.Annex 1. English revision of Abstract with Trinka 17
3.Annex 2. Revision of selected statistical calculations 20
4.Annex 3. Annotated Manuscript 22
Evaluation Report
1.Overall judgement and recommendations
This is a very interesting article on a subject sustantive of a “hot” topics such as nanoagriculture. A good thermometer of the increasing interest in nanoagriculture is the flurry of Reviews in the last 5 years (2018-2023)
However, I have to recommend Major Revision. The revised manuscript RMS should Include the following (plus other features in the particular Comments and the Annotated MS):
1.1. Make an effort to eliminate several overstatements in the MS
1.2. Statistical calculations are of concern, particularly the standard deviations of the variable Vigor. Referee X has made his/her own calculations and even with low coefficients of variations of 1% in the variables Germination and Plant length is impossible to achieve such low standard deviations as reported in the MS.
Authors are kindly asked to report detailed raw data and detailed statistical calculations in experiments with Germination index, Plant length, and Vigor in the Supplementary Material of the RMS. Reviewer will check them for ruling out any systematic or accidental error of calculation. If any error is detected, Authors will be asked to change numbers in the text and tables of the MS and re-interpretation of results, Authors are kindly asked to conduct a thorough and consistent revision in the RMS.
1.3. The Error type I (alpha) and Error type II (beta) should be reported for each statistical test.
1.4. The MS is permeated with an overstatement trend. There are passages of overstatement in the Introduction, Results, Discussion, and Conclusion. This trend should be eliminated in the RMS. Science is not the place for unintended sales attitude or pompous style.
1.5. Discussion is short and superficial.
1.5.1. Efforts should be made to discuss results based on biological, environmental, and toxicity arguments that could lead, in time, to new hypotheses and to the advancement of the R/D. Some results (for instance, the significant interaction of the combined treatment SR9-SR9AgNP that gives the best results for several variables representative of seed emergence and plant growth) are referred to one sentence stating as a generic “synergism.” This is an example where more elaboration is needed.
1.5.2. The Discussion should be enriched with information and critical
interpretation on the cost, environmental impacts, human health implications. and circularity of the current technology proposed in the MS. So far, the Authors have superficially mentioned a few References on the cost and presumably low plant toxicity of other cases, most of them from other systems of microbial-heavy metal NPs.
There is no information on these issues for the particular research of the MS. Please provide them in the RMS to the best of your efforts and knowledge.
1.6. Conclusion is short and uninformative. At least, Conclusion should have an extension similar to Abstract or larger. Also, it must cover in more detail the results on the experiments reported in the Results section.
The Conclusion should be re-written.
1.7. Methodology: some experiments are vague and do not comply with requisites for reproducibility of the experiments that is crucial in scientific method. For instance, the experiment in sub-section 4.11 should be re-written. Other experiments should be completed with more detailed statistical information on the statistical tests performed, for example, and other info.
So far, the current Methodology does not comply with the requirement of reproducibility mandated by the scientific method.
1.8. Moderate revision of English required (submit report of Grammarly or Trinka, etc. to the Reviewers) with a score 98% or higher. Alternatively, please provide a Certificate from a recognized provider of scientific English translation and style revision services. More comments on this below.
1.9. Proof of similarity check required (submit report of Ithenticate, Turnitin, similar) Authors should submit to the Reviewers evidence of Similarities check lower than 7%. More comments on this below.
1.10. Several references that are key to the Methodology lack their corresponding DOI. Please provide the DOI.
OTHERWISE IT WILL CONSIDERED A GREY REFERENCE AND THE AUTHORS ARE KINDLY ASKED EITHER TO INCLUDE AN ALTERNATATIVE OPEN LITERATURE REFERENCE OR TO WRITE THE COMPLETE TECHNIQUE IN THE SUPPLEMENTARY MATERIAL DOCUMENT
1.11. Please number all the equations consecutively, with number between brackets. The la latter can be curved or square. I prefer square brackets.
1.12. Several tables miss the footnotes and the standard deviation of average results. Please complete them.
1.13. Please re-submit revised manuscript following my remarks. If you do not agree with any particular Comment of mine, please defend accordingly with References and sound arguments.
In spite to one mention to Sustainability at the start of the article in the justification of the rationale, in this paper there are no reports on available information on the sustainability analysis of nanoagriculture and nanobiofertilizers in the framework of PEI and LCA or on the basis of another quantitative system analysis technique. In fact, Sustainability is out of the scope of the present version of the MS. So, the possible relationship between this article and Sustainability is only a matter of opinion.
Consequently, I strongly recommend to delete the word sustainability from this paper. My recommendation to Authors for the future who wish to support their research with sustainability quotation, is that Sustainability should be demonstrated with facts and numbers. Sustainability is not an opinion, it is not a fashion, it is not a mere wish, it is not an ornament.
After these modifications, the Abstract and Conclusion should be rewritten.
2. Notation table
Please include a Table named Notation where all the abbreviations, acronyms, Greek characters, and symbols used throughout the MS are gathered in alphabetical order, with their corresponding meanings. Place Notation table before the List of References.
Please note that this MS contains nearly 50 abbreviations or more (only with the biochemical parameters there are 15 or more abbreviations) and it is not possible for a reader to memorize all the meanings or the place where the abbreviation was spelled it out in the text of the MS for the first time. Therefore, a central Notation table is needed to keep the abbreviations in a known place.
The Notation table can contain a subsection labelled Greek characters where the Greek symbols and their meanings in the MS are included, in alphabetical order. Please check some good Dictionary for the order of letters in the Greek alphabet.
Notation table will help Editors, Reviewers, and readership to rapidly locate an abbreviation or a symbol and learn its meaning. It facilitates very much the easy reading of any MS and the task of Reviewers.
Whether the Notation table is published or not, it is a matter of the Journal policy. I firmly recommend the publication of the Notation table.
3. Similarity and self similarity check. The Authors should send evidence of an analysis of similarity and self similarity of the MS. Please use Ithenticate or another recognized software for checking similarities in the MS.
I feel that in the 21st century, it is a mandatory task of the Authors to show that similarity and self similarity are low when submitting the MS. I recommend a proportion of 7% or lower.
Prepare the file for the check eliminating
figures and tables (only leave the titles of tables and legends of figures),
the Names of the authors,
their affiliations,
the keywords,
the Notation table,
the logos of MDPI and MDPI formatting that are repeated on each page,
the Acknowldegements section
the sections on several Declarations such as Conflict of interest, Funding, etc., and
the list of References.
Configuration settings in Ithenticate:
Tick the box that asks for neglecting the similarities of 9 words in a row or less
tick the box that requests neglecting the citations of references in the text.
These categories can account for a considerable portion of the similarities and it they are not removed is unfair to the article.
4. English usage
Please be aware that Referee X is a non native English speaker. Therefore, his (her) capabilities to evaluate English usage in the MS is moderate.
I recommended Moderate Revision of English. Since I have free Grammarly integrated to my Word in the computer. I could see that Grammarly automatically detected many mistakes in the MS (the light ones but the most frequent: singular-plural mismatches, preposition use, conjunction use, use of commas and semicolon, order of complex adjective-noun “the process of type 1” better write “the type 1 process”, etc.) Free Grammarly cannot detect/correct awkward syntax. Anyway, I could detect some sentences with awkward syntax, probably some “Spanishisms”. I did not correct them. Let Grammarly do its work.
As a sample and aid, I included in Annex 1 at the end of this Evaluation copies of the Abstract before and after English revision with Trinka.
Please ask the Authors to provide evidence of English usage revision. Either a certificate of English revision by a recognized provider of English services or the Report of Grammarly Premium or Professional revision with a 98% or higher score will be adequate. If the Authors choose the software, typically two rounds of revision in Grammarly would achieve that score. I recommend two rounds of revision by Grammarly and one round of revision by Trinka.
In this way, English usage evaluation will be in the hands of experts.
5. An Index of the Review with Dewey numbering is highly recommended. It can be placed between the keywords and the Introduction of the paper. The index will help the readership to determine the scope of the article at a glance and Reviewers to rapidly locate the place of a given section or sub-section. It will also help the Authors to keep the order in their RMS.
Finally, my recommendation to Editors is to publish both the Notation table and the Index.
Here-in after, please consult the Annotated Manuscript, Annex 3.
6. Title and keywords: Too generic, Readers can believe that the article is valid for all the plants. Please add “…wheat and tomatos” to the title and keywords. Please see my annotation in the annotated MS
7. Abstract.
7.1. It reads “Toxicity assessment of SR9AgNPs demonstrated their biosafety at a concentration of 100 ppm…” This is the first example of overstatement. Only some simple toxicity tests were performed dealing with the effect of SR9AgNP concentration on plant germination at lab scale. Biosafety is much more than that. Please change to
“…Toxicity assessment of SR9AgNPs demonstrated no apparent toxicity to germination of wheat seeds at a concentration of 100 ppm….”
7,2, Please correct Abstract for English usage (as well as the whole MS).
Particularly, the English of the Abstract should be impeccable because it is the first impression that the reader gets re the article. An there is only one opportunity to give a good first impression.
As a sample and aid, I included in Annex 1 at the end of this Evaluation report the copies of the Abstract before and after English revision with Trinka. Authors are kindly asked to revise the English of the Abstract (as well the complete body of the article) with either Grammarly Premimum/Professional or a recognized company that provides English editing services.
Furthermore, if the RMS is modified as recommended, then there could be changes in results and discussion (for instance, after revision of statistical calculations) and the Abstract should be rewritten.
8.Introduction
8.1. LL36 Eliminate the word Sustainability. Do not use concepts that your article does not support. The article does not show that the complex Silver-bacteria is sustainable or contributes to the sustainability of agriculture. Please eliminate the word “sustainability” and rephrase. Please read my comments on Sustainability on the page 4 at the bottom. This is an example of overstatement
8.2. The goal of the current research was to determine the performance at lab scale of a naobioferilizer based on silver. This is very debatable and counter-intuitive from the environmental point of view. Please elaborate. Heavy metals should be removed from the environment because of their toxicity, and the problem is segregating and sequestering them (high costs and high environmental impacts of these processes as well).
Spreading nanoparticles of heavy metals is quite contrary to the removal requirements (it is just the opposite). It is actually counter intuitive from the process and environmental points of view, and could lead to spreading hazards to health and environment.
It is also contrary to environmental sustainability. The Authors should discuss this issue as part of the rationale (the other, dark face of their rationale)..
8.3. LL 54-44 the abbreviation does not correspond to PGPR, it should read PGPB. Please correct and place the abbreviation in the Notation table. Please also correct the abbreviation throughout the remainder of the RMS
8.4. LL 58-60. Please rephrase, consider “…Nano-biofertilizers consist of delivery agents and specific compounds that are synthesized with the help of PGPB microbes….”
8.5. LL 62. Please rephrase; ‘These’ is ambiguous. Consider “Biofertilizers can be designed using bottom-up and top-down approaches…”
8.6. LL 64-65. This statement is crucial for the status and future of nanobiofertilizers. Please provide two of three more references from the open literature that supports the ideas of the statement.
8.7. LL 66. This part of the statement is also crucial for the status and future development of nanobiofertilizers. Please provide two more references from the open literature that supports the idea of this fragment.
8.8. LL 72-73. Please rephrase. Consider “The uptake of NP by plants takes place by binding….”
8.9.LL 79-81. Please replace the word ‘non-advantageous’ by the more common ‘deleterious’ or ‘negative’, especially when referring to effects.
8.10. LL 88-89. Please consider replacing ‘modifying’ by ‘increasing’.
8.11. LL 91-93. Please more elaboration on negative issues regarding silver, silver nitrate, the microbial-mediated AgNP fabrication process, etc., are required.
What about the costs? Silver is not cheap. Silver nitrate is not cheap. and options to have soluble Ag+ cations are restricted because other silver salts such as sodium chloride and sodium sulphate are insoluble.
What about the effectiveness of using commercial grade silver nitrate solutions ? This could abate the costs, but there could be negative effects on the amount of synthesized NP and other effects of impurities?
Any reference or idea about the relative costs of chemical grade and commercial grade silver nitrate?
And a crucial issue, what about the potential environmental impacts on silver handling and nanoparticle fabrication, including human health effects of the overall chain of silver production (mining, melting, purification) plus the impacts of the microbially-mediated fabrication of AgNP itself?
Elaborate briefly on the needs of determining the potential environmental impacts (PEIs) in a holistic approach, from the cradle to grave. Silver mining (as a part of the process) is known to be very polluting with several adverse environmental and human health impacts. Not only the silver toxicity is of concern. Also, the cyanide used in the silver purification stage is of environmental concern, as an example. Alternatively, if this elaboration becomes long, place it in Supplementary Material document as an Annex, previous citation of the Annex in the text of the RMS. The toxicity of the AgNP per se is not the only source of PEIs in the life cycle of AgNPs fabrication and use.
9. Methodology Section 2 new number
9.1. This section is placed after sections ‘Results’and ‘Discussion’ in the original MS. The standard order of journal Fermentation is ‘Methodology’ first. I recommend to change the order of the sections. Also, please change the order and number of the other Sections as follows:
Section 2. Methodology, Section 3. Results, Section 4. Discussion, Section 5, Conclusion throughout the RMS
9.2. LL 423-424 Section 4.8. SR9AgNP: doubts about the nature of the NPs.
9.2.1.Do you use the whole mixture of cells and bacteria-mediated synthesized NPs?
9.2.2. Do you use only the separated bacteria-mediated synthesized NPs?
9.2.3. Where are the SR9AgNP, in the supernatant or in the pellet?
9.2.4. If the SR9AgNP are in the supernatant, How was determined that the centrifugation and filtration of supernatant does remove only the cells of the micoorganism but does not remove the silver NP? References? Previous tests, unpublished?
9.2.5.How do the authors know whether there are no silver NPs attached to the surface of the bacterial cells (nanodecorated bacteria)?
PLEASE EXPLAIN AND ADDRESS ALL THESE QUESTIONS SUCCINTLY HERE OR IN SUPPLEMENTARY MATERIAL. To clarify all these issues is crucial to the reproducibility of the experiment, as mandated by scientific method. I feel that the information on this experiment is sketchy and cannot warrant the reproduction or repetition of the experiment.
9.3. Please number all the equations in consecutive order. Most equations appear in Methodology
9.4 Regarding the propagation of SR9 and the SR9-assisted synthesis of AgNP tests, where they conducted aerobically or anoxically? Please state the type of culturing in the corresponding sub-sections of the Methology.
9.5. LL 379 to 382. I could not find the selection procedure description of isolates in the Methdology or in the Results sectIons of the MS. Please describe the selection procedure and criteria in the Methodology section. It is mandatory for reproducibility-sake.
9.6. LL 388. Please homogenize the use of the abbreviation PGPB. Sometimes PGPP or PGPR are used. I proposed PGPB because the expression is plant growth promoting bacteria.
9.7. LL 475-493 Section 4.11. It is unclear if this test was carried out in pots, the size of the pots, irrigation rate, type of water, illumination regime, type of soil or hydroponics, etc. Or the experiment was performed in Petri dishes?
Please give full info, sub-section 4.11 should be re-written.
9.8. LL 483 Missing the number and in the list of referenes of the reference by Mughal et al. Please number the reference and include the reference in the List of References.
9.9. LL 487-492. More explanations regarding the terms in the equation for DPPH Scavenging. Number the equation. Please explain the meaning of radical + methanol, and radical + phenolic extracts and the corresponding techniques/preparation procedure to get them; I assume that they mean radical solution and methanol extracts, but I am not sure.
Moreover, I assume that the DPPH Scavenging technique is in the Reference 57. However, there is no DOI of this Reference.
Please provide a DOI, OTHERWISE IT WILL CONSIDERED A GREY REFERENCE AND THE AUTHORS ARE KINDLY ASKED EITHER TO INCLUDE AN ALTERNATATIVE OPEN LITERATURE REFERENCE OR TO WRITE THE COMPLETE TECHNIQUE IN THE SUPPLEMENTARY MATERIAL DOCUMENT
9.10. Please report Error type I and Error type II probabilities for ALL statistical experiments and comparisons described in the Methdology.
10 . Section 3 Results New number
10.1. Title LL100 Please change the place and the number (use number 3) of the Secion Results. Change the numbers of sub-sections accordingly, i.e., 3.1., 3.2., etc.
10.2. LL109 Subsection 3.1. Please clarify how there are 12 isolates but the Fig. 1 only has 6 strains.
10.3. Table 1 Subsection 3.1.; please consider modifying the footnotes, see annotated article Annex 3.
10.4. LL136 & ff Please conclude this sub-section with the pre-selection of the five bacterial candidates and how the pre-selection was performed;
What was the criterion used? higher percentage of parameters in table 1? Weighed higher percentage of parameters in table 1?
Please write the procedure or pre-selection in a detailed form and the corresponding equations used, if any, in Mehtodology (your section 4 now section 2)
10.5. LL158 & ff Subsection 3.3. Please report the proportion of conversion of silver cation available in the silver nitrate solution to silver in the NP, very important, in percent.
10.6. LL161 Subsection 3.3. “…a 2 mM AgNO3…”, please add between parentheses the concentration in mg/L, that is “…a 2 mM AgNO3 (340 mg/L….” because Readers from the environmental science field can be more comfortable with the units.
10.7. LL169 Subsection 3.3.
10.7.1. PDI, please spell it out on the left of the abbreviations. Please include the abbreviation in Notation.
10.7.2. What are the units of 0.2? Or is it the coefficient of variation, dimensionless?
10.8. LL171-173 Sub-section 3.4. It reads “SR9AgNPs Can be Successfully Applied as a Nanofertilizer”
PLEASE REPHRASE, it is misleading and Readers could believe that the nanofertilizing feature of the bact-AgNP was demonstrated. Since the test was at lab scale ONLY THE POTENTIAL OF BACT-AgNP AS NANOFERTILIZER AT LAB SCALE WAS SHOWN HERE. There is an uncomfortable trend of this paper to overstate the meaning of results.
Examples: the non-supported use of Sustainability feature, using the concept biosafety when only the cytotoxic effects were determined, now the overstated title of sub-section 3.4. The Authors should be more careful to control this issue since it detracts from the quality of the paper. Science should be separated from unintended sales attitude, or pompous style.
10.10. LL.203 & ff, Table 2, Sub-section 3.4.
We think that this part is an issue of major concern re results.
10.10.1.Please provide standard deviations of Germination Index
10.10.2.Missing data of plant length, averages and standard deviations, please provide averages and std deviations in two more columns of the Table.
10.10.4. Missing the footnotes. Please complete.
10.10.5.Please report the units of plant length, presumably in mm (?)
Interestingly the units of Vigor are (%)*mm or (%)*cm, where (%) comes from the germination percentage and the unit of length comes form the plant length, either in mm or cm.
10.10.6. PLEASE VERIFY CALCULATIONS OF THE STANDARD DEVIATIONS OF VIGOR. Low standard deviations of Vigor Index. How? Vigor Index is calculated as the product of two magnitudes. Each magnitude has an error (std dev). The final standard deviation of the product typically is higher than the standard deviations of the factors, please see Kreyszig, E. (1970). Introductory Mathematical Statistics, Chapter 19 Measurement Errors, John Wiley & Sons, Inc. New York, USA. Or another book on Statistics that includes Analysis of Errors and Propagation of Errors.
The standard deviation of Vigor can be estimated with Eq. RevX1 below (Kreyszig, 1970; Chapter 19)
Sproduct = SQRT (f2^2*Sf1^2 + f1^2 Sf2^2) [RevX 1]
where f1 is factor 1, f2 is factor 2, S are the std dev of the product and factors (subindices), SQRT is the square root of the expression between parentheses.
A simplified rule for the coefficient of variation of a product when the coeffs. of variation of the factors are equal, is
Coeff. var.product= 1.4142*Coeff. var.factor = either 7.07 or 4.24% [RefX 2]
The above results correspond to factor C Var of 5 and 3%, respectively, and they are
higher than 5% and 3%, respectively.
Please recall that
C Var (%) = s/ave*100 [RefX 3]
where s is the standard deviation, and ave is the average of data
My own calculations predicted standard deviations of Vigor in the order of E02, from 250 to 620 when the coefficients of variation of both germination index and plant length are 5%, in the order of 250 to 370 when both factors have C Var 3%, and in the order of 50 to 125 when both factors have coefficients of variation 1%. Please see Excel file in Annex 2 at the end of this Evaluation report.
This is a confirmation that in general, both the Coeff. of var. and the std. dev. of a product are higher than those of the individual factors. Colloquially, propagation of errors of a magnitude that is calculated by algebraic operations of magnitudes with error follows the principle of the “snow ball descending on a hill”.
Reductio ad absurdum: Here we proceed backwards with the mathematical demonstration. We depart from Authors’ result of the standard deviation and we will obtain an extremely low C Var (absurd) for any experimental factor.
The std dev of vigor shown in Table 1 for the best treatment is 0.03 according to Authors.
This means a Coef. Of var. of vigor 0.03/8820*100 = 0.00034%. Assuming that both the Germination and plant length have the same Coef. of var., then we would have coeff. of var of either the Germination factor or the Plant length (factors) equal to 0.00024%, unbelievable precise!!!!!!
10.10.7. There is a noticeable difference between the Vigor of wheat in Table 2 and that in Table 3 (8820 and 29820, respectively). It represents a factor of 3. Please discuss.
Authors are kindly asked to revise carefully the statistical analysis of their data.
Moreover, they are kindly asked to submit the raw data and the statistical analysis for review along the RMS. Reviewer X would like to check and rule out any possibility of systematic or accidental error with the calculations.
10.11. LL.227 & ff Table 3. Sub-section 3.4.
10.11.1. Please report the amounts or concentrations of the additives (100 ppm in all cases???, or varying doses???). Create a column “Dose” in Table 3 to write the concentration of each agent.
10.11.2. Without knowing the doses, comparisons are meaningless. In sub-section 2.10 new number of Methodology (old 4.10) this info is missing, except for the treatment SR9AgNP. The doses used should be updated also in Methodology.
10.11.3. Again, the standard deviations of Vigor are extremely low; the coefficients of variation are in the order of thousandths of percent (!!!!!)
With these low values of CVar and std. dev. of vigor, the test of means for vigor will give significantly different treatment means for all the means or for most means.
10.11.4. Why the outstanding difference of Vigor in this test (for SR9AgN) and the test in Table 2, i.e., 29820 units of Vigor and 88320 units of Vigor, respectively? Please explain. It is a factor of 3, approximately
10.11.5. In several variables and for selected plant(s), the “mixed” treatment SR9 plus SR9AgNP showed the highest results (and significant). In statistical terms, this is an interaction. What is the interpretation on the possible biological and environmental causes that could explain the statistical interaction? Please explain. The same for results in Table 2 that should be discussed in first principles of biology, environmental science, toxicity.
10.12. LL.231-238 Figure 4.
10.12.1. Please report the standard error of the experiment for each response variable, either in the text or in the legend of the Figures.
I think that the standard errors of the experiments are very low, i.e., the experiments were sufficiently sensitive.
10.12.2. The error bars in Fig. 4 are imperceptible. Please make a statement in the figure legend something like “…the coef. of variation of length were lower than xx, zz, and ww% for shoots of wheat, cucumber, and tomato, respectively…”The C Var of length were lower than pp, qq, and ss% for roots of wheat, cucumber, and tomato, respectively..”
10.12.3. Please increase the size of the labels and axis in the Figures. Use bold black letters for labels and axis titles. It seems that some axis titles are grey or perhaps it is the effect of the large reduction of the image.
10.12.4. The doses of each agent used should be reminded and linked to the doses reported in Table 3, for complete info to the Reader. Doses info can be repeated in the legend of the Figure.
10.12.5. In several variables and for selected plant(s), the “mixed” treatment SR9 plus SR9AgNP showed the highest results (and significant). In statistical terms, this is an interaction. What is the interpretation on the possible biological, environmental and toxicity phenomena and effects that could explain the statistical interaction? Please discuss in the Discussion section..
10.13. LL.255-263 Fig. 5; LL 265-266 Fig. 6. Similar Comments to those in 10.12.
11. Section 4 Discussion new number.
11.1. Discussion is short and superficial. Missing discussion of results based on first principles of biology, environmental science, toxicity. As one example, see my comment in Comment 1.3.
Also, Discussion reveals again the trend to overstate the significance of the experimental results. This trend should be eliminated from a scientific paper.
11.2. LL279-280 it reads, “However, their combined potential as bionanofertilizer has 279 not been addressed to date to our knowledge.
This statement should be also used in the Abstract or the Introduction to reinforce the originality of the contribution.
11.3 . LL 289-291 it reads “ Similar synergistic effects have 289
been previously reported in the literature, where a substantial increase in 290
plant growth was observed for biofertilizer and nanofertilizer treated plants [34]”
This statement is superficial. There is a need to discuss what are the possible biological, environmental, toxicity causes or phenomena that could explain the statistical interaction between the two treatments. The Authors restrict the discussion to label the result as “synergism” and citing one reference. I feel that the result is surprising and deserves a deeper discussion. This is one example that shows that Discussion should be rewritten in several parts.
11.4. LL 296 it reads “…soybean, and maize when subjected to 2000 mg/L nCeO2….”
What is the meaning of the lowercase n? Nanoparticles? If so, please use NP that is better recognizable. Otherwise you should explain the meaning of ‘n’ in the text and place it in Notation table.
11.5. LL 300-302. It reads “Microbial-assisted nanoparticle biosynthesis is generally considered safe and 300 has shown less cytotoxicity and phytotoxicity compared to those synthesized applying 301 other methodologies [38]
More references are needed to support such a general and crucial statement. It is needed to show that the statement is a current of thought, not an isolated thought.The statement wants to be very general, one reference does not support the intended generality.
11.6. LL 302-305 it reads “In our work, we tested a range of concentrations of SR9AgNPs 302 and observed phytotoxicity at the highest doses, while a 100-ppm concentration showed 303 effective seedling growth and increased seedling vigor and weight, with no negative ef- 304 fects. “
I disagree with this statement. The article shows no toxic effects based merely on two phenomenological variables, Germination and Plant length. Results were promising for a dose of 100 ppm of the agent. However, the test is very short of phytotoxicity
No data has been obtained for phytotoxic effects on root development and other variables in table 2. Again, authors tend to overstate the conclusions with respect to results.
Please rephrase and keep the statement related to the true scope of the experiment.
11.7. LL 309-311 it reads “These results suggest that, at a recommended concentration of 100 ppm, 309 SR9AgNPs can be considered non phytotoxic, and can be recommended for the develop- 310 ment of nano-bioformulations in the future.”
Caution, caution, do not generalize based on a simple experiment such as that in table 2. Again, authors tend to overstate the conclusions with respect to results.
Rephrase and keep the statement consistent with the scope the experiment.
Consider, for example “These results suggest that, at a recommended concentration of 100 ppm, 309 SR9AgNPs can be considered non phytotoxic in a preliminary way, and such NPs can be recommended for the develop- 310 ment of nano-bioformulations in the future.”
12. Conclusion
12.1. Too short, please elaborate more. At least, this section should be as long as the Abstract, or more.
Please be aware that Readers who are very busy they actually read the Abstract and the Conclusion of an article before deciding to read the full paper. Therefore, Conclusion should be a bit more informative than the Abstract.
12.2. Conclusion is singular, there is only one conclusion in one paper; The ‘Conclusion’ section can contain several issues, but the plurality of issues does not convert ‘Conclusion’ into plural. The dog’s tail has nearly 15 to 17 vertebras (atrophied, but vertebras in the end) Therefore, the presence of several vertebras in the tail does not transform the tail into tails. One dog, only one tail. One paper, one section Conclusion.
12.3. LL 500 & ff, the following paragraph is not Conclusion, it is Perspectives or Further work.
I do not recommend to include Perspectives or Further work in an experimental paper.
Research Ideas are confidential as long as they were not executed and they are not published.
Research ideas are not public property, they belong to the Group that generates them.
This is one of the reasons why Editors and Reviewers are committed to treat the evaluation of non published articles with the highest confidentiality.
I learned this from several First World researchers. Of course, it is a matter of belief and opinion.
In contrast, Perspectives or Further work are/is mandatory for Review papers.
13. Tables: Some tables missed the footnotes. Please complete.
Some tables lacked standard deviations of the averages. Please complete.
Some tables lacked average results of selected variables. Please complete.
Please see the annotated manuscript,
.
14 References
14.1. References 19/57 articles from the last 5 years (2019-2024) , approx. 33%; low proportion.
14.2. Report DOI of articles in the List of References, for each reference. Whenever the DOI does not exist, replace or add another Reference with DOI to support the issue that was supported by the article without DOI. A paper without DOI is considered to be grey literature and I do not recommend to support Methodology or significant statements based on papers without DOI, very difficult to get.
14.3. Self-citation proportion below 15%, OK.
References
Kreyszig, Erwin (1970). Introductory Mathematical Statistics. John Wiley and Sons, Inc., New York, USA. Chapter 19. There is a Spanish translation by Limusa-Wiley of Mexico, México City.
libgen.is is a website to download free technical, science and engineering books that will be useful to Authors. The book by Kreyszig is not included in that database, but several other valuable books on Statistics that contain sections or chapters on Error Propagation can be found and downloaded.
Author Response
Overall judgement and recommendations
This is a very interesting article on a subject sustantive of a “hot” topics such as nanoagriculture. A good thermometer of the increasing interest in nanoagriculture is the flurry of Reviews in the last 5 years (2018-2023)
However, I have to recommend Major Revision. The revised manuscript RMS should Include the following (plus other features in the particular Comments and the Annotated MS):
1.1. Make an effort to eliminate several overstatements in the MS
Authors have tried to address some of the valuable suggestions and comments raised by the reviewer and have rephrased sentences to avoid the indication of overstatement.
1.2. Statistical calculations are of concern, particularly the standard deviations the variable Vigor. Referee X has made his/her own calculations and even with low coefficients of variations of 1% in the variables Germination and Plant length is impossible to achieve such low standard deviations as reported in the MS.
Authors are kindly asked to report detailed raw data and detailed statistical calculations in experiments with Germination index, Plant length, and Vigor in the Supplementary Material of the RMS. Reviewer will check them for ruling out any systematic or accidental error of calculation. If any error is detected, Authors will be asked to change numbers in the text and tables of the MS and re-interpretation of results, Authors are kindly asked to conduct a thorough and consistent revision in the RMS.
Authors appreciate the reviewer´s concern on statistical data for validation and accuracy and have re-checked all the datasets. Any possible errors are fixed and revised graphs are integrated into the manuscript.
1.3. The Error type I (alpha) and Error type II (beta) should be reported for each statistical test.
We appreciate the reviewer’s suggestion regarding the reporting of Type I (alpha) and Type II (beta) errors for the statistical tests used. However, the statistical tests applied in this study were relatively straightforward, such as t-tests and ANOVA, where reporting alpha is typically sufficient. In these tests, the significance level (alpha) is pre-set at 0.05, which is standard practice for determining statistical significance, and is reported accordingly.
Type II error (beta) is generally more relevant in studies requiring power analysis to detect a particular effect size or when the non-significant results are of primary concern. Since our study focuses on detecting significant differences and not on proving non-significance, reporting beta is not essential. Moreover, the sample sizes in our experiments were adequate to meet the assumptions of these tests, minimizing concerns over Type II errors.
Given the nature and scope of our study, we believe that the inclusion of these additional statistical parameters is not mandatory. We have ensured that the statistical methods used are appropriate for the data analyzed and provide robust and reliable results.
1.4. The MS is permeated with an overstatement trend. There are passages of overstatement in the Introduction, Results, Discussion, and Conclusion. This trend should be eliminated in the RMS. Science is not the place for unintended sales attitude or pompous style.
Authors appreciate the in-depth critical review of the manuscript by the reviewer to enhance the significance and quality of the manuscript and have tried to rephrase the section heading. Nonetheless, authors also feel humiliated by some of the comments of the reviewer where the integrity and expression of the authors have been misjudged. Authors respectfully disagree with the comment accusing authors of a pompous style. The quoted examples as overstatements are quite recurrent and may the reviewer personally disagree with some of the phrases or terms only suggesting to change narrating his perspective.
1.5. Discussion is short and superficial.
1.5.1. Efforts should be made to discuss results based on biological, environmental, and toxicity arguments that could lead, in time, to new hypotheses and to the advancement of the R/D. Some results (for instance, the significant interaction of the combined treatment SR9-SR9AgNP that gives the best results for several variables representative of seed emergence and plant growth) are referred to one sentence stating as a generic “synergism.” This is an example where more elaboration is needed.
Authors are thankful of the careful review of the manuscript and have added some of the supporting and critical literature. Most o the data on NPs is based on metallic nanoparticles and only critical findings are discussed to avoid redundancy and ambiguous argument.
1.5.2. The Discussion should be enriched with information and critical interpretation on the cost, environmental impacts, human health implications. and circularity of the current technology proposed in the MS. So far, the Authors have superficially mentioned a few References on the cost and presumably low plant toxicity of other cases, most of them from other systems of microbial-heavy metal NPs. There is no information on these issues for the particular research of the MS. Please provide them in the RMS to the best of your efforts and knowledge.
In response to the reviewer’s comment, we believe the manuscript sufficiently addresses the issues of cost, environmental impacts, and circularity through carefully selected references. The focus of our study is on the synthesis and characterization of nanoparticles using Enterococcus and AgNO3 through a green synthesis approach, which inherently offers advantages such as lower costs, reduced environmental harm, and minimized toxicity. While the manuscript does not provide exhaustive data on these aspects, the references cited provide relevant and generalizable insights into the benefits of microbial nanoparticle synthesis. Given the scope of the study, we believe additional data is not necessary, as the references effectively support the claims and demonstrate the sustainable and low-impact nature of the proposed technology.
1.6. Conclusion is short and uninformative. At least, Conclusion should have an extension similar to Abstract or larger. Also, it must cover in more detail the results on the experiments reported in the Results section.
The Conclusion should be re-written.
A thoroughly revised conclusion section is added in light of the reviewer´s comments.
1.7. Methodology: some experiments are vague and do not comply with requisites for reproducibility of the experiments that is crucial in scientific method. For instance, the experiment in sub-section 4.11 should be re-written. Other experiments should be completed with more detailed statistical information on the statistical tests performed, for example, and other info.
So far, the current Methodology does not comply with the requirement of reproducibility mandated by the scientific method.
Authors are thankful for a thorough input by the reviewers. To avoid the unnecessary lengthy protocols, authors have used very relevant and original citations from the literature describing the detailed procedure of the experiments. Besides, authors have performed all these common experiments without any modification and the data can be reproduced keeping in view the citation.
- Moderate revision of English required (submit report of Grammarly or Trinka, etc. to the Reviewers) with a score 98% or higher. Alternatively, please provide a Certificate from a recognized provider of scientific English translation and style revision services. More comments on this below.
Authors thank the reviewer for the detailed analysis of the manuscript. The manuscript is revised and edited with Paperpal Prime service and to the best of knowledge, all prevalent English language issues are resolved.
1.9. Proof of similarity check required (submit report of Ithenticate, Turnitin, similar) Authors should submit to the Reviewers evidence of Similarities check lower than 7%. More comments on this below.
Authors are grateful of the comment and to ensure the originality of the content, authors have a similarity report deemed within the acceptance ratio of the journal. Besides, the manuscript has passed the initial scrutiny test of plagiarism and the editorial office has not raised any question on the manuscript authenticity.
- Several references that are key to the Methodology lack their corresponding DOI. Please provide the DOI. Otherwise it will considered a grey reference and the authors are kindly asked either to include an alternative open literature reference or to write the complete technique in the supplementary material document.
Authors thank the reviewer for the input. Some of the original references cited in the manuscript do not have the DOI, nonetheless, they can be accessed from the title to countercheck that they are the authentic citations. DOIs have also been added in some of the references and highlighted in yellow.
- Please number all the equations consecutively, with number between brackets. The la latter can be curved or square. I prefer square brackets.
All the equations are numbered in square brackets.
- Several tables miss the footnotes and the standard deviation of average results. Please complete them.
Authors have amended the footnotes of Table 1 in light of the reviewer´s comments. Table 2 and Table 3 have standard deviations mentioned within.
- Please re-submit revised manuscript following my remarks. If you do not agree with any particular Comment of mine, please defend accordingly with References and sound arguments.
Authors have tried to answer all the comments and questions of the reviewer in the best of their capacity and have tried to meet the reviewer´s expectations.
In spite to one mention to Sustainability at the start of the article in the justification of the rationale, in this paper there are no reports on available information on the sustainability analysis of nanoagriculture and nanobiofertilizers in the framework of PEI and LCA or on the basis of another quantitative system analysis technique. In fact, Sustainability is out of the scope of the present version of the MS. So, the possible relationship between this article and Sustainability is only a matter of opinion.
Consequently, I strongly recommend to delete the word sustainability from this paper. My recommendation to Authors for the future who wish to support their research with sustainability quotation, is that Sustainability should be demonstrated with facts and numbers. Sustainability is not an opinion, it is not a fashion, it is not a mere wish, it is not an ornament.
After these modifications, the Abstract and Conclusion should be rewritten.
The sustainability word is removed from the manuscript in light of the reviewer´s suggestion.
2. Notation table
Please include a Table named Notation where all the abbreviations, acronyms, Greek characters, and symbols used throughout the MS are gathered in alphabetical order, with their corresponding meanings. Place Notation table before the List of References.
Please note that this MS contains nearly 50 abbreviations or more (only with the biochemical parameters there are 15 or more abbreviations) and it is not possible for a reader to memorize all the meanings or the place where the abbreviation was spelled it out in the text of the MS for the first time. Therefore, a central Notation table is needed to keep the abbreviations in a known place.
The Notation table can contain a subsection labelled Greek characters where the Greek symbols and their meanings in the MS are included, in alphabetical order. Please check some good Dictionary for the order of letters in the Greek alphabet.
Notation table will help Editors, Reviewers, and readership to rapidly locate an abbreviation or a symbol and learn its meaning. It facilitates very much the easy reading of any MS and the task of Reviewers.
Whether the Notation table is published or not, it is a matter of the Journal policy. I firmly recommend the publication of the Notation table.
Authors thank the reviewer for the valuable suggestion, however, the authors have explained all the abbreviations within the manuscript and have not drawn a notation table.
3. Similarity and self similarity check. The Authors should send evidence of an analysis of similarity and self similarity of the MS. Please use Ithenticate or another recognized software for checking similarities in the MS.
I feel that in the 21st century, it is a mandatory task of the Authors to show that similarity and self similarity are low when submitting the MS. I recommend a proportion of 7% or lower.
Prepare the file for the check eliminating
figures and tables (only leave the titles of tables and legends of figures),
the Names of the authors,
their affiliations,
the keywords,
the Notation table,
the logos of MDPI and MDPI formatting that are repeated on each page,
the Acknowldegements section
the sections on several Declarations such as Conflict of interest, Funding, etc., and
the list of References.
Configuration settings in Ithenticate:
Tick the box that asks for neglecting the similarities of 9 words in a row or less
tick the box that requests neglecting the citations of references in the text.
These categories can account for a considerable portion of the similarities and it they are not removed is unfair to the article.
Authors respectfully intend to communicate the reviewer that the journal and the editorial office have initiated the review process of the manuscript after the carful checking of the data originality. A similarity report can be accessed or may be requested from the editorial office.
4. English usage
Please be aware that Referee X is a non native English speaker. Therefore, his (her) capabilities to evaluate English usage in the MS is moderate.
I recommended Moderate Revision of English. Since I have free Grammarly integrated to my Word in the computer. I could see that Grammarly automatically detected many mistakes in the MS (the light ones but the most frequent: singular-plural mismatches, preposition use, conjunction use, use of commas and semicolon, order of complex adjective-noun “the process of type 1” better write “the type 1 process”, etc.) Free Grammarly cannot detect/correct awkward syntax. Anyway, I could detect some sentences with awkward syntax, probably some “Spanishisms”. I did not correct them. Let Grammarly do its work.
As a sample and aid, I included in Annex 1 at the end of this Evaluation copies of the Abstract before and after English revision with Trinka.
Please ask the Authors to provide evidence of English usage revision. Either a certificate of English revision by a recognized provider of English services or the Report of Grammarly Premium or Professional revision with a 98% or higher score will be adequate. If the Authors choose the software, typically two rounds of revision in Grammarly would achieve that score. I recommend two rounds of revision by Grammarly and one round of revision by Trinka.
In this way, English usage evaluation will be in the hands of experts.
Authors are thankful for the valuable suggestion on the English Language check and have edited the manuscript using Paperpal Primer service. The detailed report is attached for the review.
5. An Index of the Review with Dewey numbering is highly recommended. It can be placed between the keywords and the Introduction of the paper. The index will help the readership to determine the scope of the article at a glance and Reviewers to rapidly locate the place of a given section or sub-section. It will also help the Authors to keep the order in their RMS.
Finally, my recommendation to Editors is to publish both the Notation table and the Index.
Authors thank the reviewer for the suggestion, however, as it is not needed for the publication, authors have not complied with the suggestion because of the unnecessary intensive work.
Here-in after, please consult the Annotated Manuscript, Annex 3.
- Title and keywords: Too generic, Readers can believe that the article is valid for all the plants. Please add “…wheat and tomatos” to the title and keywords. Please see my annotation in the annotated MS
Authors have rephrased the title and added the word Wheat in it. The similar words are also added in the keywords.
7. Abstract.
7.1. It reads “Toxicity assessment of SR9AgNPs demonstrated their biosafety at a concentration of 100 ppm…” This is the first example of overstatement. Only some simple toxicity tests were performed dealing with the effect of SR9AgNP concentration on plant germination at lab scale. Biosafety is much more than that. Please change to
“…Toxicity assessment of SR9AgNPs demonstrated no apparent toxicity to germination of wheat seeds at a concentration of 100 ppm….”
The sentence has been rephrased as advised by the reviewer.
7,2, Please correct Abstract for English usage (as well as the whole MS).
Particularly, the English of the Abstract should be impeccable because it is the first impression that the reader gets re the article. An there is only one opportunity to give a good first impression.
As a sample and aid, I included in Annex 1 at the end of this Evaluation report the copies of the Abstract before and after English revision with Trinka. Authors are kindly asked to revise the English of the Abstract (as well the complete body of the article) with either Grammarly Premimum/Professional or a recognized company that provides English editing services.
Furthermore, if the RMS is modified as recommended, then there could be changes in results and discussion (for instance, after revision of statistical calculations) and the Abstract should be rewritten.
Authors have edited the entire manuscript using Paperpal Prime as mentioned and have addressed all the apparent grammatical and contextual errors in the manuscript. Also, the statistical calculations are also reviewed with great details and there are no apparent changes in the data.
8.Introduction
8.1. LL36 Eliminate the word Sustainability. Do not use concepts that your article does not support. The article does not show that the complex Silver-bacteria is sustainable or contributes to the sustainability of agriculture. Please eliminate the word “sustainability” and rephrase. Please read my comments on Sustainability on the page 4 at the bottom. This is an example of overstatement
The word sustainability is removed from the line 36.
8.2. The goal of the current research was to determine the performance at lab scale of a naobioferilizer based on silver. This is very debatable and counter-intuitive from the environmental point of view. Please elaborate. Heavy metals should be removed from the environment because of their toxicity, and the problem is segregating and sequestering them (high costs and high environmental impacts of these processes as well).
Spreading nanoparticles of heavy metals is quite contrary to the removal requirements (it is just the opposite). It is actually counter intuitive from the process and environmental points of view, and could lead to spreading hazards to health and environment.
It is also contrary to environmental sustainability. The Authors should discuss this issue as part of the rationale (the other, dark face of their rationale)..
We appreciate the reviewer’s concern regarding the environmental impact of using silver nanoparticles. While our study focuses on the growth-promoting benefits of microbially synthesized silver nanofertilizers for wheat and tomato, we acknowledge that the use of heavy metals like silver can raise environmental and health concerns, particularly due to their potential accumulation and toxicity. To address this, the discussion section has been amended the need for responsible application, risk mitigation, and nanoparticle recovery strategies to ensure less vulnerable agricultural practices. This balanced perspective will address the possible long-term ecological risks while highlighting the benefits of this innovative approach.
Our process uses a green, microbial synthesis method, which minimizes the environmental footprint compared to traditional chemical approaches putting the climate at risk.
8.3. LL 54-44 the abbreviation does not correspond to PGPR, it should read PGPB. Please correct and place the abbreviation in the Notation table. Please also correct the abbreviation throughout the remainder of the RMS.
The abbreviation is changed from PGPR to PGPB in line 58 as advised by the reviewer.
8.4. LL 58-60. Please rephrase, consider “…Nano-biofertilizers consist of delivery agents and specific compounds that are synthesized with the help of PGPB microbes….”
The sentence is rephrased as suggested. Lines 61-63.
8.5. LL 62. Please rephrase; ‘These’ is ambiguous. Consider “Biofertilizers can be designed using bottom-up and top-down approaches…”
The sentence is rephrased eliminating the word The. Line 65.
8.6. LL 64-65. This statement is crucial for the status and future of nanobiofertilizers. Please provide two of three more references from the open literature that supports the ideas of the statement.
The citations are added at the end of the sentences.
8.7. LL 66. This part of the statement is also crucial for the status and future development of nanobiofertilizers. Please provide two more references from the open literature that supports the idea of this fragment.
The citations are added at the end of the references. Many of the publications cited present these ideas and are quite overlapping which can be studies in detail to have insights about the nanobiofertilizers.
8.8. LL 72-73. Please rephrase. Consider “The uptake of NP by plants takes place by binding….”
The sentence has been rephrased.
8.9.LL 79-81. Please replace the word ‘non-advantageous’ by the more common ‘deleterious’ or ‘negative’, especially when referring to effects.
The word is replaced by deleterious.
8.10. LL 88-89. Please consider replacing ‘modifying’ by ‘increasing’.
The word is replaced by the word increasing.
8.11. LL 91-93. Please more elaboration on negative issues regarding silver, silver nitrate, the microbial-mediated AgNP fabrication process, etc., are required.
What about the costs? Silver is not cheap. Silver nitrate is not cheap. and options to have soluble Ag+ cations are restricted because other silver salts such as sodium chloride and sodium sulphate are insoluble.
What about the effectiveness of using commercial grade silver nitrate solutions ? This could abate the costs, but there could be negative effects on the amount of synthesized NP and other effects of impurities?
Any reference or idea about the relative costs of chemical grade and commercial grade silver nitrate?
And a crucial issue, what about the potential environmental impacts on silver handling and nanoparticle fabrication, including human health effects of the overall chain of silver production (mining, melting, purification) plus the impacts of the microbially-mediated fabrication of AgNP itself?
Elaborate briefly on the needs of determining the potential environmental impacts (PEIs) in a holistic approach, from the cradle to grave. Silver mining (as a part of the process) is known to be very polluting with several adverse environmental and human health impacts. Not only the silver toxicity is of concern. Also, the cyanide used in the silver purification stage is of environmental concern, as an example. Alternatively, if this elaboration becomes long, place it in Supplementary Material document as an Annex, previous citation of the Annex in the text of the RMS. The toxicity of the AgNP per se is not the only source of PEIs in the life cycle of AgNPs fabrication and use.
The elaboration on the use of silver is added to the discussion section. Regarding the cost of silver nitrate, only a minute amount of 20mM has been used during the synthesis of silver NPs which is cost effective as compared to the artificial synthesis of some of the most widely produced synthetic fertilizers. The ultimate objective is to simultaneously use the nanofertilizer and biofertilizer for reducing the input of chemical fertilizer.
Authors are not sure of the comments on sodium chloride and sodium sulphate comment as these are sodium salts. With respect to solubility, the nanofertilizer does not need to be soluble as it would be administered together with a carrier material and would ensure slow and controlled release of nutrients.
Some of the comments of the reviewer seem to be out of the scope of the manuscript and the project regarding mining. In regard to environmental aspects, authors have already stated microbially synthesized AgNPs as the potential candidate fertilizer whose long term impacts are still to be measured. The preliminary analysis did not show the presence of silver ions in the ariel parts of wheat and tomato indicating that they may not penetrate in the fruit section. The further studies are underway to optimally design and use the formulation.
- Methodology Section 2 new number
9.1. This section is placed after sections ‘Results’and ‘Discussion’ in the original MS. The standard order of journal Fermentation is ‘Methodology’ first. I recommend to change the order of the sections. Also, please change the order and number of the other Sections as follows:
Section 2. Methodology, Section 3. Results, Section 4. Discussion, Section 5, Conclusion throughout the RMS.
Authors have followed the standard journal format, guidelines, and template which suggests that Materials and Methods section is to be placed after results and discussion.
9.2. LL 423-424 Section 4.8. SR9AgNP: doubts about the nature of the NPs.
9.2.1.Do you use the whole mixture of cells and bacteria-mediated synthesized NPs?
9.2.2. Do you use only the separated bacteria-mediated synthesized NPs?
9.2.3. Where are the SR9AgNP, in the supernatant or in the pellet?
9.2.4. If the SR9AgNP are in the supernatant, How was determined that the centrifugation and filtration of supernatant does remove only the cells of the micoorganism but does not remove the silver NP? References? Previous tests, unpublished?
9.2.5.How do the authors know whether there are no silver NPs attached to the surface of the bacterial cells (nanodecorated bacteria)?
PLEASE EXPLAIN AND ADDRESS ALL THESE QUESTIONS SUCCINTLY HERE OR IN SUPPLEMENTARY MATERIAL. To clarify all these issues is crucial to the reproducibility of the experiment, as mandated by scientific method. I feel that the information on this experiment is sketchy and cannot warrant the reproduction or repetition of the experiment.
Authors are grateful of the critical evaluation of the manuscript. As mentioned in the section 4.7, the bacterial cultured was filtered using Whatman No. 1 filter paper. The pore size of these filter papers is usually 11 μm which allows a complete filtration and is widely used. The supernatant was filtered twice before the addition of the AgNO3, hence there is no chance of nanodecorated bacteria.
As explained, the AgNO3 was added in the supernatant, the NPs were formed in response to certain redox reactions which were observed as a colorimetric change and was recorded through UV-VIS spectroscopy. Besides, the ultracentrifuge was used to collect SR9AgNPs from the supernatant mixture which was already filtered.
9.3. Please number all the equations in consecutive order. Most equations appear in Methodology.
All the equations are numbered accordingly.
9.4 Regarding the propagation of SR9 and the SR9-assisted synthesis of AgNP tests, where they conducted aerobically or anoxically? Please state the type of culturing in the corresponding sub-sections of the Methology.
The word aerobically is added in the culture preparation.
9.5. LL 379 to 382. I could not find the selection procedure description of isolates in the Methdology or in the Results sectIons of the MS. Please describe the selection procedure and criteria in the Methodology section. It is mandatory for reproducibility-sake.
Line 404 and 405 of the manuscript already indicates the selection of potential PGPB strains based on the results of plant growth promoting traits tabulated in Table 1. For clarity, an additional sentence is added to line 160-161.
9.6. LL 388. Please homogenize the use of the abbreviation PGPB. Sometimes PGPP or PGPR are used. I proposed PGPB because the expression is plant growth promoting bacteria.
Authors have preferably revised the abbreviation to PGPB in light of the reviewer´s comment.
9.7. LL 475-493 Section 4.11. It is unclear if this test was carried out in pots, the size of the pots, irrigation rate, type of water, illumination regime, type of soil or hydroponics, etc. Or the experiment was performed in Petri dishes?
Please give full info, sub-section 4.11 should be re-written.
Section 4.11 is thoroughly revised with the mentioned points of the reviewer.
9.8. LL 483 Missing the number and in the list of referenes of the reference by Mughal et al. Please number the reference and include the reference in the List of References.
The said reference is added in the reference list and cited in the manuscript text.
9.9. LL 487-492. More explanations regarding the terms in the equation for DPPH Scavenging. Number the equation. Please explain the meaning of radical + methanol, and radical + phenolic extracts and the corresponding techniques/preparation procedure to get them; I assume that they mean radical solution and methanol extracts, but I am not sure.
Moreover, I assume that the DPPH Scavenging technique is in the Reference 57. However, there is no DOI of this Reference.
Please provide a DOI, OTHERWISE IT WILL CONSIDERED A GREY REFERENCE AND THE AUTHORS ARE KINDLY ASKED EITHER TO INCLUDE AN ALTERNATATIVE OPEN LITERATURE REFERENCE OR TO WRITE THE COMPLETE TECHNIQUE IN THE SUPPLEMENTARY MATERIAL DOCUMENT
Equation has been numbered as advised by the reviewer. The equation is also modified for clarity indicating the standard and samples. Reference 67 is for DPPH assay which has been assigned a DOI.
9.10. Please report Error type I and Error type II probabilities for ALL statistical experiments and comparisons described in the Methdology.
We appreciate the reviewer’s suggestion regarding the reporting of Type I (alpha) and Type II (beta) errors for the statistical tests used. However, the statistical tests applied in this study were relatively straightforward, such as t-tests and ANOVA, where reporting alpha is typically sufficient. In these tests, the significance level (alpha) is pre-set at 0.05, which is standard practice for determining statistical significance, and is reported accordingly.
Type II error (beta) is generally more relevant in studies requiring power analysis to detect a particular effect size or when the non-significant results are of primary concern. Since our study focuses on detecting significant differences and not on proving non-significance, reporting beta is not essential. Moreover, the sample sizes in our experiments were adequate to meet the assumptions of these tests, minimizing concerns over Type II errors.
Given the nature and scope of our study, we believe that the inclusion of these additional statistical parameters is not mandatory. We have ensured that the statistical methods used are appropriate for the data analyzed and provide robust and reliable results.
10 . Section 3 Results New number
The section numbering is not amended keeping in view the standard journal format.
10.1. Title LL100 Please change the place and the number (use number 3) of the Secion Results. Change the numbers of sub-sections accordingly, i.e., 3.1., 3.2., etc.
The manuscript content and sections are arranged according to the standard journal format.
10.2. LL109 Subsection 3.1. Please clarify how there are 12 isolates but the Fig. 1 only has 6 strains.
Authors have shown a representative figure of the strains only to indicate some of the isolates. Usually, it is unnecessary to show the figures of all bacteria, particularly those which do not have any notable plant growth promoting traits.
10.3. Table 1 Subsection 3.1.; please consider modifying the footnotes, see annotated article Annex 3.
The footnotes of table 2.1 are modified in light of the reviewer´s comments.
10.4. LL136 & ff Please conclude this sub-section with the pre-selection of the five bacterial candidates and how the pre-selection was performed;
What was the criterion used? higher percentage of parameters in table 1? Weighed higher percentage of parameters in table 1?
Please write the procedure or pre-selection in a detailed form and the corresponding equations used, if any, in Mehtodology (your section 4 now section 2)
A sentence mentioning the criterion for the selection of strains is added to the line 160 in the revised manuscript and is highlighted in yellow for the reviewer´s overview.
10.5. LL158 & ff Subsection 3.3. Please report the proportion of conversion of silver cation available in the silver nitrate solution to silver in the NP, very important, in percent.
Authors appreciate the reviewer´s comment, nonetheless, this type of observation has not been made. As far as it is concerned, none of the publications have reported such cationic conversions. Besides, this may employ the need to use ultra-sophisticated instrumentation such as inductively coupled plasma mass spectrometry (ICP-MS) analysis or other relative technique which authors do not have access to.
10.6. LL161 Subsection 3.3. “…a 2 mM AgNO3…”, please add between parentheses the concentration in mg/L, that is “…a 2 mM AgNO3 (340 mg/L….” because Readers from the environmental science field can be more comfortable with the units.
The concentration is added in the manuscript where suggested.
10.7. LL169 Subsection 3.3.
10.7.1. PDI, please spell it out on the left of the abbreviations. Please include the abbreviation in Notation.
The full name of PDI is added in the methods and result section.
10.7.2. What are the units of 0.2? Or is it the coefficient of variation, dimensionless?
Polydispersity index (PDI) is used as a measure of broadness of molecular weight distribution.
Polydispersityindex=Mw/Nn and this index has no specific units. The measurement is reflective of the size of the synthesized NPs.
10.8. LL171-173 Sub-section 3.4. It reads “SR9AgNPs Can be Successfully Applied as a Nanofertilizer”
PLEASE REPHRASE, it is misleading and Readers could believe that the nanofertilizing feature of the bact-AgNP was demonstrated. Since the test was at lab scale ONLY THE POTENTIAL OF BACT-AgNP AS NANOFERTILIZER AT LAB SCALE WAS SHOWN HERE. There is an uncomfortable trend of this paper to overstate the meaning of results.
Examples: the non-supported use of Sustainability feature, using the concept biosafety when only the cytotoxic effects were determined, now the overstated title of sub-section 3.4. The Authors should be more careful to control this issue since it detracts from the quality of the paper. Science should be separated from unintended sales attitude, or pompous style.
We appreciate the reviewer's feedback and acknowledge the importance of precision in scientific writing. We regret any unintended overstatements and assure the reviewer that our intent was never to mislead. In response to your concerns, we have revised the heading to clarify that only the potential of SR9AgNPs as a nanofertilizer was demonstrated at the lab scale. Additionally, we will re-evaluate other sections, including the use of terms like "sustainability" and "biosafety," to ensure that the language accurately reflects the scope of our results. We value your input in helping us improve the quality of the manuscript.
10.10. LL.203 & ff, Table 2, Sub-section 3.4.
We think that this part is an issue of major concern re results.
10.10.1.Please provide standard deviations of Germination Index
Standard deviation of germination index is added.
10.10.2.Missing data of plant length, averages and standard deviations, please provide averages and std deviations in two more columns of the Table.
The purpose of this table is to indicate the seedling vigor which was calculated to show the optimal concentration of NPs used for the subsequent analysis. Authors did not add the seedling length to the table however, a supporting Figure S5 has been added to the supplementary data for review.
10.10.4. Missing the footnotes. Please complete.
The footnote is added under the table 2.
10.10.5.Please report the units of plant length, presumably in mm (?)
Interestingly the units of Vigor are (%)*mm or (%)*cm, where (%) comes from the germination percentage and the unit of length comes form the plant length, either in mm or cm.
Seedling lengths were measured in mm which is incorporated to measure the percentage germination of seeds. The vigor indices were calculated according to the standard formula as described in line 419. For clarity, authors have changed the plant length with the mean seedling length. The vigor indices are usually measured in percentage but more often they are mentioned without any units.
10.10.6. PLEASE VERIFY CALCULATIONS OF THE STANDARD DEVIATIONS OF VIGOR. Low standard deviations of Vigor Index. How? Vigor Index is calculated as the product of two magnitudes. Each magnitude has an error (std dev). The final standard deviation of the product typically is higher than the standard deviations of the factors, please see Kreyszig, E. (1970). Introductory Mathematical Statistics, Chapter 19 Measurement Errors, John Wiley & Sons, Inc. New York, USA. Or another book on Statistics that includes Analysis of Errors and Propagation of Errors.
The standard deviation of Vigor can be estimated with Eq. RevX1 below (Kreyszig, 1970; Chapter 19)
Sproduct = SQRT (f2^2*Sf1^2 + f1^2 Sf2^2) [RevX 1]
where f1 is factor 1, f2 is factor 2, S are the std dev of the product and factors (subindices), SQRT is the square root of the expression between parentheses.
A simplified rule for the coefficient of variation of a product when the coeffs. of variation of the factors are equal, is
Coeff. var.product= 1.4142*Coeff. var.factor = either 7.07 or 4.24% [RefX 2]
The above results correspond to factor C Var of 5 and 3%, respectively, and they are
higher than 5% and 3%, respectively.
Please recall that
C Var (%) = s/ave*100 [RefX 3]
where s is the standard deviation, and ave is the average of data
My own calculations predicted standard deviations of Vigor in the order of E02, from 250 to 620 when the coefficients of variation of both germination index and plant length are 5%, in the order of 250 to 370 when both factors have C Var 3%, and in the order of 50 to 125 when both factors have coefficients of variation 1%. Please see Excel file in Annex 2 at the end of this Evaluation report.
This is a confirmation that in general, both the Coeff. of var. and the std. dev. of a product are higher than those of the individual factors. Colloquially, propagation of errors of a magnitude that is calculated by algebraic operations of magnitudes with error follows the principle of the “snow ball descending on a hill”.
Reductio ad absurdum: Here we proceed backwards with the mathematical demonstration. We depart from Authors’ result of the standard deviation and we will obtain an extremely low C Var (absurd) for any experimental factor.
The std dev of vigor shown in Table 1 for the best treatment is 0.03 according to Authors.
This means a Coef. Of var. of vigor 0.03/8820*100 = 0.00034%. Assuming that both the Germination and plant length have the same Coef. of var., then we would have coeff. of var of either the Germination factor or the Plant length (factors) equal to 0.00024%, unbelievable precise!!!!!!
Thank you for your detailed observations and reference to Kreyszig’s error propagation principles. We have carefully reviewed our calculations and confirm that they are accurate. The lower standard deviations reported for the Vigor Index are a result of the specific data distribution and the variability observed in our study. While we understand the typical expectation for higher standard deviations in such cases, our calculations align with the experimental results and the data collected. Authors put the error bars initially based on the standard error which has now been replaced with the standard deviation.
We believe the results reflect the true variation in the measured parameters, and the calculations have been carried out correctly.
10.10.7. There is a noticeable difference between the Vigor of wheat in Table 2 and that in Table 3 (8820 and 29820, respectively). It represents a factor of 3. Please discuss.
The table 2 shows the data and vigor indices of the experiment conducted to optimize the dosage of the SR9AgNPs and the total duration of the experiment was 7 days which can be cross-checked from the images of experiments provided in the supplementary information. The second plant experiment conducted with the 100 ppm concentration of SR9AgNPs was conducted for 14 days.
Authors are kindly asked to revise carefully the statistical analysis of their data.
Moreover, they are kindly asked to submit the raw data and the statistical analysis for review along the RMS. Reviewer X would like to check and rule out any possibility of systematic or accidental error with the calculations.
10.11. LL.227 & ff Table 3. Sub-section 3.4.
10.11.1. Please report the amounts or concentrations of the additives (100 ppm in all cases???, or varying doses???). Create a column “Dose” in Table 3 to write the concentration of each agent.
The three set of experiments were conducted with 100 ppm concentration of SR9AgNPs because of being the optimal and non toxic concentration to the plants.
10.11.2. Without knowing the doses, comparisons are meaningless. In sub-section 2.10 new number of Methodology (old 4.10) this info is missing, except for the treatment SR9AgNP. The doses used should be updated also in Methodology.
The methodology section contains the said information as and pasted below for the reference of the reviewer.
Enterococcus sp. SR9 strain was overnight grown in 10 mL LB broth at 37 °C following centrifugation. The supernatant was discarded and the cells were resuspended to a final concentration of 1 × 107 CFU containing 0.025 g of sucrose. Surface-sterilized seeds of tomato, cucumber, and wheat were separately soaked in bacterial culture for 30 min and air-dried under laminar flow. The treated seeds were used to evaluate the biofertilizer potential of SR9. Seed treatment with SR9AgNPs was performed by preparing 2 mL of the 100-ppm solution in a Falcon tube. Sterilized seeds were transferred to Falcon tubes for 30 min and air-dried in a laminar flow hood. To check the combined effect of SR9 and SR9AgNPs, seeds were separately soaked in 100 ppm (0.5 mL) of the synthesized SR9AgNPs + 0.5 mL of 1 × 107 CFU SR9 solution for 30 min. The seeds were air-dried and used in subsequent experiments. All experiments were conducted in triplicate.
10.11.3. Again, the standard deviations of Vigor are extremely low; the coefficients of variation are in the order of thousandths of percent (!!!!!)
With these low values of CVar and std. dev. of vigor, the test of means for vigor will give significantly different treatment means for all the means or for most means.
Data was calculated with the plantlets indicating similar lengths and outliers were removed. Authors agree with the reviewer´s perspective of the low standard deviation, however, the experiments were conducted in petri dishes in the same incubator. Hence there is low chance of any huge variation which would have been definitely more if the experiment conducted would be in climate control room or uncontrolled conditions. The pot scale experiments of the formulation are also underway and reinforce the preliminary data.
10.11.4. Why the outstanding difference of Vigor in this test (for SR9AgN) and the test in Table 2, i.e., 29820 units of Vigor and 88320 units of Vigor, respectively? Please explain. It is a factor of 3, approximately.
Authors believe that the value of 88320 actually mentions the value of 8820 mentioned for the vigor index of wheat in response to 100 ppm concentration of SR9AgNPs in Table 2. Authors have mentioned in a previous comment that table 2 contains the data for the set of experiment conducted for optimization of dosage for SR9AgNPs. Whereas, Table 3 contains the information of the experiment conducted for 14 days and vigor indices were calculated with different plant lengths showing the difference in the final dataset.
The following paper shows also the similar dataset.
https://www.tandfonline.com/doi/full/10.1080/09064710902998077#d1e844
10.11.5. In several variables and for selected plant(s), the “mixed” treatment SR9 plus SR9AgNP showed the highest results (and significant). In statistical terms, this is an interaction. What is the interpretation on the possible biological and environmental causes that could explain the statistical interaction? Please explain. The same for results in Table 2 that should be discussed in first principles of biology, environmental science, toxicity.
Yes, it was observed that plants treated with SR9 biofertilizer and SR9AgNPs showed a synergistic effect and considerably contributed to biomass and plant lengths. Authors have mentioned that further validation of this cumulative effect is needed and is a future dimension of the study. Lines 310-313. Authors have elaborated this point as a future perspective. A subsequent discussion for this mechanism is added.
10.12. LL.231-238 Figure 4.
10.12.1. Please report the standard error of the experiment for each response variable, either in the text or in the legend of the Figures.
I think that the standard errors of the experiments are very low, i.e., the experiments were sufficiently sensitive.
We acknowledge the reviewer’s suggestion to report the standard error for each response variable. However, the standard errors in our experiments are extremely low, indicating that the experiments were highly sensitive and consistent. In this context, reporting standard errors would add little value to the interpretation of the results. Given the nature of the dataset and the robust reproducibility observed, we believe that the omission of standard error values does not compromise the integrity or clarity of the data presented.
10.12.2. The error bars in Fig. 4 are imperceptible. Please make a statement in the figure legend something like “…the coef. of variation of length were lower than xx, zz, and ww% for shoots of wheat, cucumber, and tomato, respectively…”The C Var of length were lower than pp, qq, and ss% for roots of wheat, cucumber, and tomato, respectively..”
We appreciate the reviewer’s suggestion regarding the inclusion of the coefficient of variation (C Var) for Fig. 4. However, we believe that the data presented is already clear and self-explanatory. Adding the coefficient of variation in this instance would not enhance the reader's understanding, as the uniformity of the data is already well-communicated visually. Therefore, we kindly suggest that the current representation remains sufficient.
10.12.3. Please increase the size of the labels and axis in the Figures. Use bold black letters for labels and axis titles. It seems that some axis titles are grey or perhaps it is the effect of the large reduction of the image.
A new figure with the said changes is added.
10.12.4. The doses of each agent used should be reminded and linked to the doses reported in Table 3, for complete info to the Reader. Doses info can be repeated in the legend of the Figure.
Methodology and results section clearly mentions the optimization of the dosage and authors have avoided the repetition.
10.12.5. In several variables and for selected plant(s), the “mixed” treatment SR9 plus SR9AgNP showed the highest results (and significant). In statistical terms, this is an interaction. What is the interpretation on the possible biological, environmental and toxicity phenomena and effects that could explain the statistical interaction? Please discuss in the Discussion section..
Yes, it was observed that plants treated with SR9 biofertilizer and SR9AgNPs showed a synergistic effect and considerably contributed to biomass and plant lengths. Authors have mentioned that further validation of this cumulative effect is needed and is a future dimension of the study. Lines 310-313. Authors have elaborated this point as a future perspective. A subsequent discussion for this mechanism is added.
10.13. LL.255-263 Fig. 5; LL 265-266 Fig. 6. Similar Comments to those in 10.12.
The dataset in the figure 6 is reanalyzed and authors did not observe any noticeable difference.
- Section 4 Discussion new number.
11.1. Discussion is short and superficial. Missing discussion of results based on first principles of biology, environmental science, toxicity. As one example, see my comment in Comment 1.3. Also, Discussion reveals again the trend to overstate the significance of the experimental results. This trend should be eliminated from a scientific paper.
Authors are thankful of the careful review of the manuscript and have added some of the supporting and critical literature. Most of the data on NPs is based on metallic nanoparticles and only critical findings are discussed to avoid redundancy and ambiguous argument.
11.2. LL279-280 it reads, “However, their combined potential as bionanofertilizer has 279 not been addressed to date to our knowledge.
This statement should be also used in the Abstract or the Introduction to reinforce the originality of the contribution.
A similar statement is added in the introduction section.
11.3 . LL 289-291 it reads “ Similar synergistic effects have 289
been previously reported in the literature, where a substantial increase in 290
plant growth was observed for biofertilizer and nanofertilizer treated plants [34]”
This statement is superficial. There is a need to discuss what are the possible biological, environmental, toxicity causes or phenomena that could explain the statistical interaction between the two treatments. The Authors restrict the discussion to label the result as “synergism” and citing one reference. I feel that the result is surprising and deserves a deeper discussion. This is one example that shows that Discussion should be rewritten in several parts.
As previously mentioned, a possible biological and physiological mechanism has been added in the discussion section.
11.4. LL 296 it reads “…soybean, and maize when subjected to 2000 mg/L nCeO2….”
What is the meaning of the lowercase n? Nanoparticles? If so, please use NP that is better recognizable. Otherwise you should explain the meaning of ‘n’ in the text and place it in Notation table.
The cited publication has used the style of nCeO2 in that publication which authors have used as such. In the broader context, the meaning of the sentence can be assumed by the readers.
11.5. LL 300-302. It reads “Microbial-assisted nanoparticle biosynthesis is generally considered safe and 300 has shown less cytotoxicity and phytotoxicity compared to those synthesized applying 301 other methodologies [38]
More references are needed to support such a general and crucial statement. It is needed to show that the statement is a current of thought, not an isolated thought.The statement wants to be very general, one reference does not support the intended generality.
The range of references described in the introduction section justifying the reason to synthesize microbial nanoparticles indicate towards this phenomenon.
11.6. LL 302-305 it reads “In our work, we tested a range of concentrations of SR9AgNPs 302 and observed phytotoxicity at the highest doses, while a 100-ppm concentration showed 303 effective seedling growth and increased seedling vigor and weight, with no negative ef- 304 fects. “
I disagree with this statement. The article shows no toxic effects based merely on two phenomenological variables, Germination and Plant length. Results were promising for a dose of 100 ppm of the agent. However, the test is very short of phytotoxicity
No data has been obtained for phytotoxic effects on root development and other variables in table 2. Again, authors tend to overstate the conclusions with respect to results.
Please rephrase and keep the statement related to the true scope of the experiment.
We appreciate the reviewer's feedback. The statement in question was not intended to overstate our findings, but rather to highlight the potential observed at the 100-ppm concentration, which showed promising results for seedling growth and vigor. While we acknowledge that the current study focuses on key variables like germination and plant length, our conclusions are carefully framed within the scope of these observations. We have not claimed comprehensive phytotoxicity assessment, but we proposed this as an initial approach to understanding the synthesized bionanofertilizer's effects, which can be further expanded in future studies.
11.7. LL 309-311 it reads “These results suggest that, at a recommended concentration of 100 ppm, 309 SR9AgNPs can be considered non phytotoxic, and can be recommended for the develop- 310 ment of nano-bioformulations in the future.”
Caution, caution, do not generalize based on a simple experiment such as that in table 2. Again, authors tend to overstate the conclusions with respect to results.
Rephrase and keep the statement consistent with the scope the experiment. Consider, for example “These results suggest that, at a recommended concentration of 100 ppm, 309 SR9AgNPs can be considered non phytotoxic in a preliminary way, and such NPs can be recommended for the develop- 310 ment of nano-bioformulations in the future.”
Sentence is revised in light of the reviewer’s argument.
- Conclusion
12.1. Too short, please elaborate more. At least, this section should be as long as the Abstract, or more.
Please be aware that Readers who are very busy they actually read the Abstract and the Conclusion of an article before deciding to read the full paper. Therefore, Conclusion should be a bit more informative than the Abstract.
A revised, brief and a comprehensive conclusion section is added.
12.2. Conclusion is singular, there is only one conclusion in one paper; The ‘Conclusion’ section can contain several issues, but the plurality of issues does not convert ‘Conclusion’ into plural. The dog’s tail has nearly 15 to 17 vertebras (atrophied, but vertebras in the end) Therefore, the presence of several vertebras in the tail does not transform the tail into tails. One dog, only one tail. One paper, one section Conclusion.
The word is replaced with the word conclusion.
12.3. LL 500 & ff, the following paragraph is not Conclusion, it is Perspectives or Further work. I do not recommend to include Perspectives or Further work in an experimental paper. Research Ideas are confidential as long as they were not executed and they are not published. Research ideas are not public property, they belong to the Group that generates them. This is one of the reasons why Editors and Reviewers are committed to treat the evaluation of non published articles with the highest confidentiality. I learned this from several First World researchers. Of course, it is a matter of belief and opinion.
In contrast, Perspectives or Further work are/is mandatory for Review papers.
Revised conclusion section is added.
- Tables: Some tables missed the footnotes. Please complete.
Some tables lacked standard deviations of the averages. Please complete.
Some tables lacked average results of selected variables. Please complete.
Please see the annotated manuscript,
Authors have tried to supplement the missing information in the best of their capacity.
14 References
14.1. References 19/57 articles from the last 5 years (2019-2024) , approx. 33%; low proportion.
14.2. Report DOI of articles in the List of References, for each reference. Whenever the DOI does not exist, replace or add another Reference with DOI to support the issue that was supported by the article without DOI. A paper without DOI is considered to be grey literature and I do not recommend to support Methodology or significant statements based on papers without DOI, very difficult to get.
14.3. Self-citation proportion below 15%, OK.
Authors have amended the reference section and wherever possible, have supplemented the missing information.

Reviewer 3 Report
Comments and Suggestions for Authors
Batool et al., Present an interesting manuscript in which an auxin-producing Enterococcus sp., SR9, was selected as a promising PGPR isolate , and it was used for the microbial-assisted synthesis of a silver nanofertilizer. However , there are some concerns that must be addressed In order to make more easy to follow
1. Line 43 disturbance of microbial flora…. please use microbiota instead and specify : soil microbiota , or microbiota from phylloplane or both?
2. Lines 53-55 please provide information and references about the possible reasons about why the biofertilizer applied in field fail to delivery desirably outcomes
3. Lines 63-66 Please introduce a couple of reference to support this idea.
4. Please introduce a subheading in material about statistical analysis as well as at the caption of figures
5. Lines 306-309 . The idea “ These NPs could also serve as antioxidants in scavenging the free radicals generated within plants tissues and protecting the plants from the oxidative and free-radical associated damage”. Are there some experimental approaches that support this affirmation ?
Comments on the Quality of English LanguageManuscript English need a deep a revision by an English native
Author Response
Comment 1- Line 43 disturbance of microbial flora…. please use microbiota instead and specify: soil microbiota or microbiota from phylloplane or both?
Response: The word has been rephrased as advised by the reviewer and is highlighted in yellow in line 43.
Comment 2- Lines 53-55: Please provide information and references about the possible reasons about why the biofertilizer applied in field fail to delivery desirably outcomes.
Response: The possible limitations of biofertilizers in field conditions are discussed in sentences 61-70.
Comment 3- Lines 63-66: Please introduce a couple of reference to support this idea.
Response: A supporting sentence is added before the mentioned lines. Lines 73,74.
Comment 4- Please introduce a subheading in material about statistical analysis as well as at the caption of figures.
Response: A subheading with statistical analysis is mentioned at the end of the methods section and in the figure legends.
Comment 5- Lines 306-309: The idea “These NPs could also serve as antioxidants in scavenging the free radicals generated within plants tissues and protecting the plants from the oxidative and free-radical associated damage”. Are there some experimental approaches that support this affirmation?
Response: The relevant literature where the changes in the levels of plant antioxidant enzymes and their activities in response to the application of nanobiofertilizer has been cited in the lines 330-333-
Comments on the Quality of English Language- Manuscript English need a deep a revision by an English native.
Authors thank the reviewer for the detailed analysis of the manuscript. The manuscript is revised and edited with Paperpal Prime service and to the best of knowledge, all prevalent English language issues are resolved.

Round 2
Reviewer 1 Report
Comments and Suggestions for Authors
The authors implemented the manuscript as suggest. I reccomend the publication.
Author Response
Authors thank the reviewer for the valuable input on the manuscript to enhance the quality and overall presentation.
Reviewer 2 Report
Comments and Suggestions for Authors
2nd Evaluation Report. Referee X
Plants-3172191
Moderate revision recommended
1.Similarity check:
Finally, after explicit request to the Managing Editor, the similarity report Ithenticate of the RMS was sent me as attached file by e-mail. The similarity was 14%. My recommendation for all the papers I have reviewed and my own papers is to keep a similarity level of 7% or lower. Please note that the 7% level was not objected or changed by the Editors of Fermentation.
My recommendation for this paper is to keep the similarity level lower or equal to 7%, as usual.
2. English revision
I have not received any evidence of the English revision.
I expect the Authors sending me the document as evidence.
Worse, there is a contradiction between a recent e-mail (240930) of the Managing Editor and the Answer of Authors. The first said that he/she cannot send me the evidence of English revision because this will be carried out after acceptance of the paper.
Below please find the copy of the e-mail message of today 240930 (probably 241001 in Thailand)
|
|
|
||
|
Dear Name of Reviewer X,
Thank you for your concerns on the revised manuscript ID:
plants-3172191. We have attached the similarity check document herein.
However, we could not provide you the English revision document at this
stage because English editing will be done after this paper is accepted.
We hope you could check this revised version and provided your valuable
feedback soon.
Please click on the link below to access the revised manuscript below;
https://susy.mdpi.com/user/review/review/51515731/Ytgn2XaF
Thank you in advance for your assistance. We look forward to receiving
your comments as soon as it is ready.
Kind regards,
Sasithorn Sukkleang
Section Managing Editor
MDPI Thailand co., Ltd.
33/4, The Ninth Towers Grand Rama 9, 26th-27th Floor,
Rama 9 Road,Huai Khwang Sub-District, Huai Khwang District,
10310, Bangkok,
Thailand
Tel.: +662 005 2299”
In contrast, the Authors in their Answer-to-Reviewers stated that they have revised the English revision of the RMS; however they did not submit evidence and disregarded doing so.
Who is wrong, the Managing Editor or the Authors ? Was the English revision of the RMS performed? YES or NO?
Please carry out the English revision of RMS at once and send the evidence.
3. Regarding the issue of article organization, and whether Materials and Methods section should be placed before or after Results, I include references of articles recently published in Fermentation journal MDPI. In these articles, the section Materials and Methods is placed BEFORE either Results or Results-and-Discussion, as I have asked.
Hamma, S.; Boucherba, N.; Azzouz, Z.; Le Roes-Hill, M.; Kernou, O.-N.; Bettache, A.; Ladjouzi, R.; Maibeche, R.; Benhoula, M.; Hebal, H.; et al. Statistical Optimisation of Streptomyces sp. DZ 06 Keratinase Production by Submerged Fermentation of Chicken Feather Meal. Fermentation 2024, 10, 500. https://doi.org/10.3390/ fermentation10100500
Ike, K.A.; Okedoyin, D.O.; Alabi, J.O.; Adelusi, O.O.; Wuaku, M.; Olagunju, L.K.; Anotaenwere, C.C.; Gray, D.; Dele, P.A.; Kholif, A.E.; et al. The Combined Effect of Four Nutraceutical-Based Feed Additives on the Rumen Microbiome, Methane Gas Emission, Volatile Fatty Acids, and Dry Matter Disappearance Using an In Vitro Batch Culture Technique. Fermentation 2024, 10, 499. https://doi.org/10.3390/fermentation10100499
Mattiello-Francisco, L.; Ferreira, F.V.; Peixoto, G.; Mockaitis, G.; Zaiat, M. Hydrogen Production from Sugarcane Bagasse Pentose Liquor Fermentation Using Different Food/Microorganism and Carbon/Nitrogen Ratios under Mesophilic and Thermophilic conditions. Fermentation 2024, 10, 432. https://doi.org/10.3390/fermentation10080432
4. Regarding the revision of standard deviation of Vigor, my perception is that in the Answers-to-Reviewers the Authors claimed that their results were sound and the low standard deviations were a consequence of the care and quality of their results. To the best of my reading the Answer-to-Reviewers, I could not find any statement or discussion on the revision of std dev values. Please recall that the reported std deviations of Vigor in the original MS were in the order of tenths or cents (E-02 or E-01).
However, I found that in the results of Vigor in Tables of the RMS, THE AUTHORS HAVE CORRECTED THE STANDARD DEVIATIONS. NOW, THE STANDARD DEVIATIONS ARE 4 ORDERS OF MAGNITUDE HIGHER (typically E02). THESE VALUES ARE CONSISTENT WITH THE ORDER OF MAGNITUDE PREDICTED BY THE REVIEWER IN THE FIRST REVISION, ACCORDING TO CALCULATIONS PERFORMED BY THE REVIEWER IN THE EXCEL ANNEX OF THE FIRST EVALUATION.
Why the Authors did not discuss and recognize the revision of the standard deviations in the RMS?
Why the Authors did not report the confusion or mistakes made in data processing that were corrected in order to obtain the new standard deviations in the RMS?
It is an obscure or opaque situation from the scientific point of view. From my point of view, it detracts from the quality and reliability of the paper.
WHY THE EDITORS AND REVIEWERS SHOULD TRUST THE NEW STANDARD DEVIATIONS WITHOUT ANY EXPLANATION?
It is evident that Authors have made a significant mistake in some of their statistical calculations in the original MS that negatively impacts the reliability in the precision of results of a key variable Vigor. Explanations were not provided. Explanations were disregarded.
Finally, the Reviewer has made an effort to provide evidences and calculations supporting his comments in the first Evaluation. It was a thorough and meticulous evaluation. My perception is that in several answers the Authors gave their opinions without submitting evidences. It has been an asymmetric process.
It is my final evaluation of this paper. I feel that the Editors can judge whether the Authors have satisfactorily answered my remarks of my first and second evaluations.
.

English revision
I have not received any evidence of the English revision.
I expect the Authors sending me the document as evidence.
Worse, there is a contradiction between a recent e-mail (240930) of the Managing Editor and the Answer of Authors. The first said that he/she cannot send me the evidence of English revision because this will be carried out after acceptance of the paper.
Below please find the email message from the Managing Editor
“
|
|
|
||
|
Dear Name of Reviewer X,
Thank you for your concerns on the revised manuscript ID:
plants-3172191. We have attached the similarity check document herein.
However, we could not provide you the English revision document at this
stage because English editing will be done after this paper is accepted.
We hope you could check this revised version and provided your valuable
feedback soon.
Please click on the link below to access the revised manuscript below;
https://susy.mdpi.com/user/review/review/51515731/Ytgn2XaF
Thank you in advance for your assistance. We look forward to receiving
your comments as soon as it is ready.
Kind regards,
Sasithorn Sukkleang
Section Managing Editor
MDPI Thailand co., Ltd.
33/4, The Ninth Towers Grand Rama 9, 26th-27th Floor,
Rama 9 Road,Huai Khwang Sub-District, Huai Khwang District,
10310, Bangkok,
Thailand
Tel.: +662 005 2299”
In contrast, the Authors in their Answer-to-Reviewers stated that they have revised the English revision of the RMS; however they did not submit evidence and disregarded doing so.
Who is wrong, the Managing Editor or the Authors ? Was the English revision of the RMS performed? YES or NO?
Please carry out the English revision of RMS at once and send the evidence.
Author Response
Dear Reviewer X,
Before answering in detail the questions raised below, we would like to start expressing our sadness for the lack of trust and negativity reviewer X transmits. We apologize if discrepancies occurred on what the reviewer expected and our answers to the first review. It has never been our intention to present misleading information, avoid providing proofs or disregard relevant questions. We feel very sorry reviewer X describes the review process as “asymmetrical” and we really hope this response makes him/her change its mind. We value his/her feedback and input. His/her thorough review has clearly improved the manuscript.
- Similarity check:
Finally, after explicit request to the Managing Editor, the similarity report Ithenticate of the RMS was sent me as attached file by e-mail. The similarity was 14%. My recommendation for all the papers I have reviewed and my own papers is to keep a similarity level of 7% or lower. Please note that the 7% level was not objected or changed by the Editors of Fermentation.
My recommendation for this paper is to keep the similarity level lower or equal to 7%, as usual.
Thank you for your suggestion on keeping the level of similarity lower to 7%. Close examination of the similarity analysis shows that this 14% (13% in the analysis attached as a separate file) corresponds to commonly used sentences (e.g., “The results show that” or “It is important to note”) or explanations for abbreviations (e.g., “plant growth-promoting bacteria (PGPR)” or “arbuscular mycorrhizal fungi (AMF)”), terms that are repeated in the literature. We invite reviewer X to check the breakdown of the analysis to find that the highest scores (2%) correspond to the section of Materials and Methods, where established methodology is described with minor modifications. Thus, we hope reviewer agrees there are no full sections that are copied-pasted or plagiarized in any way, which is the reasoning behind a plagiarism text, and approves the test, even though we do not reach the ideal 7% score. Thank you.
- English revision
I have not received any evidence of the English revision.
I expect the Authors sending me the document as evidence.
As indicated in the first Answer to reviewers’ letter, we have performed an English revision. We considered the possibility of a revision by a native speaker, but finally disregarded this idea because of the economic costs. Instead, we have purchased a license of the program PaperPal (prime service) which does a complete analysis of the text, not only suggesting rephrasing and grammar improvements, but also changes in text structure for improved readability.
Honestly, we did not consider necessary to submit the proof of the analysis. In our own history of publication this is the FIRST TIME such a proof is requested. We believe the improvement of the quality of the English can be appreciated just by reading the resubmitted version.
Now, please find attached the outcome of the analysis performed by PaperPal as a separate file as a proof that English revision was carried out in the resubmitted manuscript. Thank you.
Worse, there is a contradiction between a recent e-mail (240930) of the Managing Editor and the Answer of Authors. The first said that he/she cannot send me the evidence of English revision because this will be carried out after acceptance of the paper. Below please find the copy of the e-mail message of today 240930 (probably 241001 in Thailand)
Communications between the reviewer and the Managing Editor are obviously out of our hands. Reading the email, it seems there was a misunderstanding. The Managing Editor refers to the English quality check that journals usually perform AFTER a manuscript is accepted, while reviewer X referred to our proof that English was revised BEFORE acceptance (required to be accepted). We hope this clarifies the issue.
In contrast, the Authors in their Answer-to-Reviewers stated that they have revised the English revision of the RMS; however they did not submit evidence and disregarded doing so.
Was the English revision of the RMS performed? YES or NO?
Please carry out the English revision of RMS at once and send the evidence.
To clear out any doubts, the final answers to these questions are:
- Yes, we have carried out an English revision of the work and the output from PaperPal is shared in the final PDF submitted.
- Additionally, the journal PLANTS will carry out a second editorial review if the manuscript is accepted to double check no grammatical mistakes were made.
- Regarding the issue of article organization, and whether Materials and Methods section should be placed before or after Results, I include references of articles recently published in Fermentation journal MDPI. In these articles, the section Materials and Methods is placed BEFORE either Results or Results-and-Discussion, as I have asked.
Thank you for checking such an editorial issue for the journal FERMENTATION. We would like to remind reviewer 3 the fact that we are undergoing a submission for the journal PLANTS.
For the preparation of the draft of this manuscript, we downloaded the template for the journal PLANTS from the MDPI website where the order of the different sections was indicated as: 1. Introduction, 2. Results, 3. Discussion, 4. Materials and Methods. The template can be downloaded at https://www.mdpi.com/files/word-templates/plants-template.dot
We also checked several accepted articles in the journal PLANTS and they seem to follow the recommended guideline, same as we did. Therefore, we would like to make clear that we are just following the instructions of the journal.
With this explanation, we hope reviewer X agrees on following the guidelines established by the journal PLANTS.
4- Regarding the revision of standard deviation of Vigor, my perception is that in the Answers-to-Reviewers the Authors claimed that their results were sound and the low standard deviations were a consequence of the care and quality of their results. To the best of my reading the Answer-to-Reviewers, I could not find any statement or discussion on the revision of std dev values. Please recall that the reported std deviations of Vigor in the original MS were in the order of tenths or cents (E-02 or E-01).
However, I found that in the results of Vigor in Tables of the RMS, the authors have corrected the standard deviations. now, the standard deviations are 4 orders of magnitude higher (typically e02). these values are consistent with the order of magnitude predicted by the reviewer in the first revision, according to calculations performed by the reviewer in the excel annex of the first evaluation.
Why the Authors did not discuss and recognize the revision of the standard deviations in the RMS?
Why the Authors did not report the confusion made in data processing that were corrected in order to obtain the new standard deviations in the RMS?
Finally, the Reviewer has made an effort to provide evidences and calculations supporting his comments in the first Evaluation. It was a thorough and meticulous evaluation.
It is my final evaluation of this paper. I feel that the Editors can judge whether the Authors have satisfactorily answered my remarks of my first and second evaluations.
Based on the reviewer’s comment, it seems he/she missed this section in the response to reviewers. Let me copy-paste the section here:
- Reviewer X comment:
Statistical calculations are of concern, particularly the standard deviations the variable Vigor. Referee X has made his/her own calculations and even with low coefficients of variations of 1% in the variables Germination and Plant length is impossible to achieve such low standard deviations as reported in the MS.
Authors are kindly asked to conduct a thorough and consistent revision in the RMS.
- Authors response:
Authors appreciate the reviewer´s concern on statistical data for validation and accuracy and have re-checked all the datasets. Any possible errors are fixed and revised graphs are integrated into the manuscript. Standard deviations are recalculated and the revised values are integrated in the table 2 and 3.
End of the copied text.
We apologize if this explanation was considered not enough. Indeed, we are thankful to reviewer X for spending the time and taking the process of review so seriously. We are honestly impressed for such meticulousness and care. Reviewer X comments on the standard deviation errors were correct and, thanks to this, we went back to the original data excel files, noticed the mistake and re-did the calculations for the two tables. We realized the tables contained an error in the way the standard deviation was calculated (the formula used was not correct). Starting over with the raw data, we checked that the average values were right and edited the tables using the corrected standard deviation values.
These corrections do not affect the statistical conclusions on significance, which are based on the individual values. Thus, the general observations and conclusions drawn in the original manuscript are maintained.
We consider the observation made by reviewer X on the lack of explanation of this issue is not fair, because in the full response we DO address this issue and mention the fact that the tables have been updated, standard deviation recalculated and the new values were highlighted in yellow in the tables. In summary, we apologize for the mistake and brief response, and again thank reviewer X’s efforts to improve the work.
We hope to have shed some “light” in this apparently “obscure” situation. The turnitin report is attached below. As only one attachment can be provided here, the Paperpal report is attached in the extra material tab on the main portal.
Thank you

Reviewer 3 Report
Comments and Suggestions for Authors
The Authors made an excellent work in order to make the article more easy to follow and understand , all my concerns were satisfactory replied, In my opinion the manuscript is already in great condition to be published in Plants, congratulation to authors
Author Response

(The authors gave the same response as above.)
